# Suppressing electrolyte-lithium metal reactivity via Li$^+$-desolvation in uniform nano-porous separator

Li Sheng[1], Qianqian Wang[1], Xiang Liu [2], Hao Cui[1], Xiaolin Wang[1], Yulong Xu[1], Zonglong Li[1], Li Wang[1], Zonghai Chen [2], Gui-Liang Xu [2], Jianlong Wang [1], Yaping Tang[1], Khalil Amine [2,3], Hong Xu [1✉] & Xiangming He [1✉]

Lithium reactivity with electrolytes leads to their continuous consumption and dendrite growth, which constitute major obstacles to harnessing the tremendous energy of lithium-metal anode in a reversible manner. Considerable attention has been focused on inhibiting dendrite via interface and electrolyte engineering, while admitting electrolyte-lithium metal reactivity as a thermodynamic inevitability. Here, we report the effective suppression of such reactivity through a nano-porous separator. Calculation assisted by diversified characterizations reveals that the separator partially desolvates Li$^+$ in confinement created by its uniform nanopores, and deactivates solvents for electrochemical reduction before Li$^0$-deposition occurs. The consequence of such deactivation is realizing dendrite-free lithium-metal electrode, which even retaining its metallic lustre after long-term cycling in both Li-symmetric cell and high-voltage Li-metal battery with LiNi$_{0.6}$Mn$_{0.2}$Co$_{0.2}$O$_2$ as cathode. The discovery that a nano-structured separator alters both bulk and interfacial behaviors of electrolytes points us toward a new direction to harness lithium-metal as the most promising anode.

[1] Institute of Nuclear and New Energy Technology, Tsinghua University, 100084 Beijing, P. R. China. [2] Chemical Sciences and Engineering Division, Argonne National Laboratory, Lemont, IL 60439, USA. [3] Materials Science and Engineering, Stanford University, Stanford, CA 94305, USA. ✉email: hongxu@tsinghua.edu.cn; hexm@tsinghua.edu.cn

lithium-metal (Li⁰) possesses the lowest reduction potential and the smallest atomic weight among all metals on the Periodic Table, and has been considered the most prominent anode material for rechargeable batteries[1,2]. However, attempts to harness Li⁰ have been prevented by its extreme reactivity with electrolytes, especially those based on carbonate solvents, which leads to low Coulombic efficiency (CE) and dangerous morphologies such as dendritic and dead Li⁰ [3,4]. The effort to circumvent Li⁰-reversibility had directly led us to lithium-ion batteries (LIBs), which employ graphite as a Li⁺-intercalation host to replace Li⁰, at the expense of capacity and energy[4]. Two decades after the commercial success of LIBs[5,6], the pursuit for higher energy density eventually forced us back to Li⁰ [7–9]. Numerous approaches were explored to prevent dendritic and dead Li⁰ [10–12], with varying progresses achieved[10,13–17]. These efforts aim to mitigate the uneven electrodeposition of Li⁰ either chemically via electrolyte formulation so that a better interphase could be formed, or mechanically via surface/bulk engineering of the electrode to create a physical barrier against the dendritic or dead Li⁰ growth. No attempt has been made to directly resolve the fundamental reason of Li⁰-irreversibility: its extreme reactivity with electrolytes.

Here, we report the direct suppression of reactivity between Li⁰ and carbonate electrolyte via partial desolvation of Li⁺ in nanopores of a nano-structured photoresist membrane. We found that the nano-structure of polymerized photoresist can effectively regulate the energy state of Li⁺ by partially removing solvent molecules from its primary solvation sheath, enabling the reduction of Li⁺ more preferential than those solvent molecules remaining in the Li⁰-solvation sheath. The Li-Li symmetric cell or high voltage Li-metal cell with high nickel $LiNi_{0.6}Mn_{0.2}Co_{0.2}O_2$ as cathode using such nano-structured photoresist as separator membrane is shown to stabilize Li⁰ in the presence of conventional carbonate-based electrolytes, where the Li⁰ is able to retain metallic lustre even after long term-cycling, in complete absence of either dendritic or dead Li⁰. The uniform nanopores-stabilization of Li⁰ with electrolyte is so effective that such protection could be precisely patterned into spatial selective separator membrane on lithium surface via a simple contact lithographic technique. This precise spatial control could enable, among other possible applications, the printable microsized but ultra-high-energy lithium-metal batteries on circuit board substrates, which would be otherwise impossible for the extremely reactive Li⁰.

## Results and discussion
### Polymerized photoresist separator.
A uniform nano-porous membrane is synthesized from a multi-vinyl functionalized cluster (Supplementary Fig. 1), the metal-organic cluster (denoted Zr-MOC hereafter) consisting of a $Zr_6O_4(OH)_4$ core with 12 methacrylic acid (MAA) ligands (Fig. 1a). Zr-MOC has a size of ~1.6 nm (from single-crystal X-ray diffraction, Fig. 1b; CCDC No: 2022033, Supplementary Fig. 2 and Supplementary Tables 1–5); spin-coating of Zr-MOC solution onto a silicon-wafer produces an extremely smooth film (Supplementary Fig. 3). Due to the small particle size and multi-vinyl functionalized shell, Zr-MOC could serve as a photoresist, and precise patterns have been achieved under 365 nm UV exposure (Supplementary Fig. 4). Further, high-resolution patterns can also be obtained using electron-beam lithography (Supplementary Fig. 5), suggesting that Zr-MOC's polymerization could be controlled at nano-scales. Aided by these unique features, we prepared a uniform and porous membrane by photopolymerizing Zr-MOC under 365 nm UV (with a photo-sensitizer 2,4,6-trimethylbenzoyldiphenyl phosphine oxide). The highly crosslinked networks introduces nano-structure into the polymerized Zr-MOC (denoted Zr-MOCN hereafter), as

evidenced by a high Brunauer–Emmett–Teller (BET) surface area of $171 \, m^2 \, g^{-1}$ (Supplementary Figs. 6, 7) with a pore size distributed between 1.41 and 2.77 nm (Supplementary Fig. 8), which are comparable in size with a solvated Li⁺ (~1.0 nm when solvated by four carbonate molecules). In contrast, the Zr-MOC showed a BET surface area of only $9 \, m^2 \, g^{-1}$ (Supplementary Fig. 9), indicating almost no pores exist in the cluster.

To use the Zr-MOCN as separator for coin cell assembling, we fabricated a supporting membrane with stretched porous polypropylene[18] (Celgard2500, which is commonly used as separator for LIBs; denoted as PP hereafter), in which the embedded in-situ photopolymerization of Zr-MOC creates Zr-MOC network (denoted as Zr-MOCN@PP) in the percolating pores of PP separator (Fig. 1c). Visually different from the pristine PP separator (Fig. 1d, inset photo), Zr-MOCN@PP appears to be a transparent membrane (Fig. 1f, inset photo), due to its continuous structure on the microscopic level. From scanning electron microscope (SEM) and atomic force microscope (AFM) images, the Zr-MOCN@PP demonstrated an extremely smooth surface with the $R_q$ of 1.58 nm and $R_{max}$ of 12.5 nm (Fig. 1f, j, k); no stretched pores or gaps were found both from the top view and cross-section view (Fig. 1f, g), in sharp contrast with the PP separator (Fig. 1d, e; h, i). Meanwhile, Zr-MOCN@PP can be fully bent, the flexibility of which is enough for assembling coin cells.

### Performance of the photoresist separator.
To demonstrate the generality of the host membrane, we used a commercial LIB electrolyte based on lithium hexafluorophosphate (LiPF₆) dissolved in the mixture of ethylene carbonate, dimethyl carbonate and ethyl methyl carbonate (denoted LiPF₆-LE hereafter) for Li⁰ stability and electrodeposition evaluation. Both Zr-MOCN@PP and PP separators were filled with the LiPF₆-LE and sandwiched between two Li metal foils. Zr-MOCN@PP membrane exhibited apparently reduced voltage polarization (around 10 mV) than PP (80 mV) in the initial several cycling (Fig. 2a) at a current density of $1 \, mA \, cm^{-2}$ (the areal capacity was $1 \, mAh \, cm^{-2}$). Such cell could be cycled for more than 2000 h with the voltage profiles remaining essentially unchanged. Such almost identical voltage profiles detected in Zr-MOCN@PP cell over the period of 2000 h indicate a very unusual inertness, which is, to the best of our knowledge, rare observed with Li⁰.

In sharp contrast, the cell containing LiPF₆-LE in PP displays a cell polarization increased largely (around 400 mV) that significantly deteriorates, and the cell terminates with an obvious short circuit at the 270th cycle (Fig. 2a), as lithium dendrite growth can be easily identified (Fig. 2c, d and Supplementary Fig. 10). A thick black layer can also be visually identified on the surface of lithium foils (Fig. 2c, inset photo). Moreover, the cross-section SEM image (Supplementary Fig. 11a,b) revealed a thick and porous surface layer that was generated due to the constant reaction between Li⁰ and electrolyte components[19–21], which are typical for Li⁰ cycled in electrolytes that it reacts with.

However, the post-mortem analysis on Li⁰ recovered from the Zr-MOCN@PP cell reveals the extremely rare view of Li⁰ maintaining its pristine state with metallic lustre (Fig. 2e, inset photo), which strongly indicate the absence of either parasitic reactions or dendritic and dead Li⁰ (Fig. 2e, f). A dense cross-section of recovered Li⁰ further confirmed the smooth electrodeposition with negligible corrosion caused by parasitic reactions (Supplementary Fig. 11c, d). From electrochemical impedance spectroscopy (EIS) (Supplementary Fig. 12), before cycling (after 1 h of cell assembly), the Zr-MOCN@PP based cell demonstrated a slightly higher Ohmic resistance ($R_\Omega$), but a significant lower SEI resistance ($R_{SEI}$) than the pristine PP separator based cell.

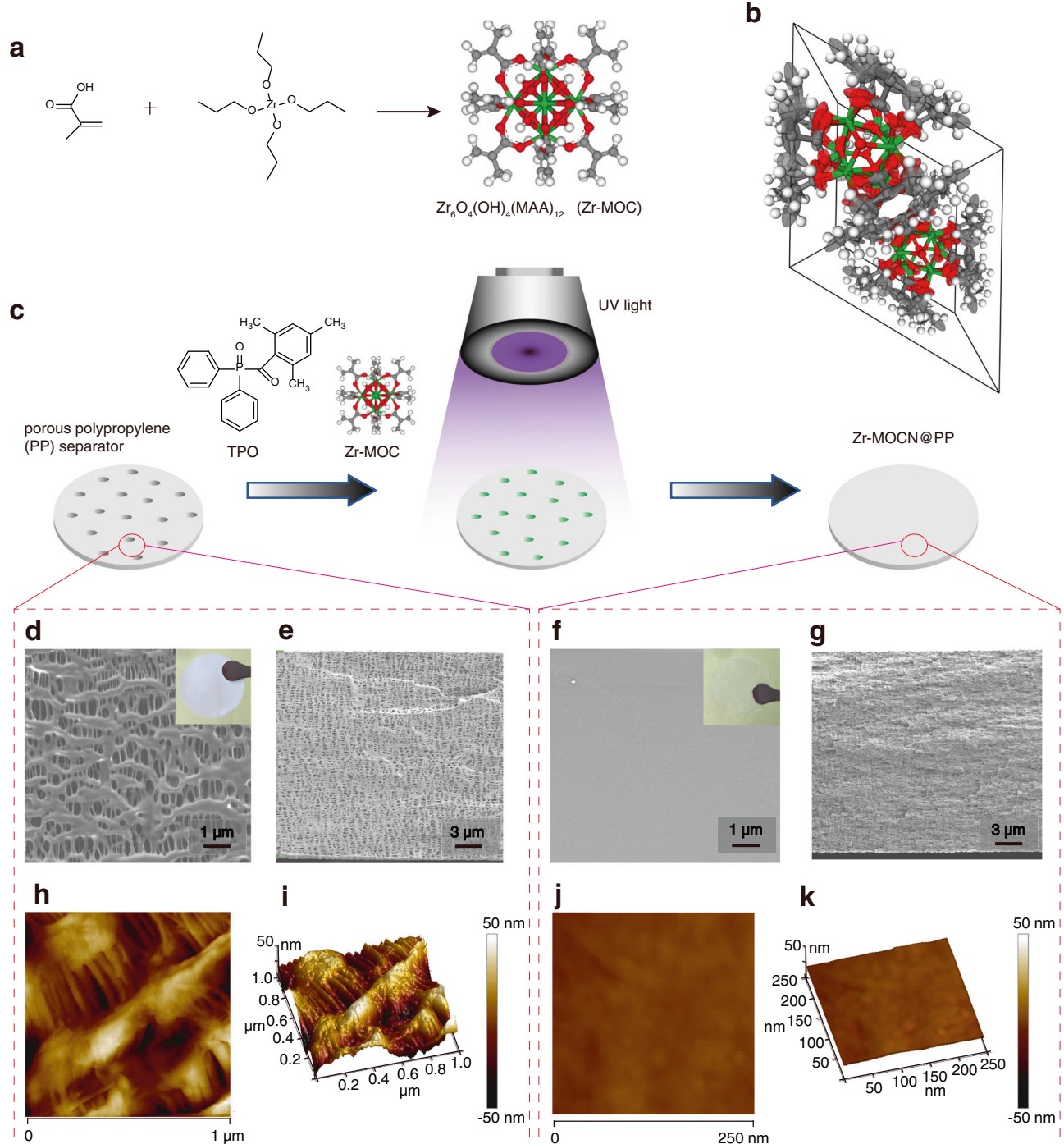

**Fig. 1 Photopolymerization of the photoresist separator and its microscopic characterization. a**, **b** Synthesis and crystalline structure of Zr-MOC. (Color code: gray ball, C; red ball, O; white ball, H; green ball, Zr). **c** Fabrication of the Zr-MOCN@PP membrane via photopolymerization. SEM images of the top and cross-section view of: **d**, **e** The PP separator; **f**, **g** Zr-MOCN@PP. Insets are photos of the membrane. AFM images of: **h**, **i** The PP separator; **j**, **k** Zr-MOCN@PP.

And for the cycled cells, the $R_{SEI}$ of Zr-MOCN@PP kept at a low value, while the $R_{SEI}$ of PP largely increased from 10th cycle to 200th cycle. Furthermore, even under an extreme situation of excessively high current density ($10\ mA\ cm^{-2}/10\ mAh\ cm^{-2}$), the Zr-MOCN@PP cell also showed a very low polarization voltage (Fig. 2b) and dendrite-free Li-deposition morphology (Supplementary Fig. 13). The parasitic reaction products were also visually negligible, as confirmed by detailed chemical analyses below. Although the suppression of the lithium dendrite by nano-

porous separator have been reported[16,22–26], inhibition the continuously occurred parasitic reaction between $Li^0$ and liquid electrolyte was firstly studied in this work.

To differentiate whether the reduced $Li^0$ reactivity arises from the Zr-MOC (pristine $Zr_6O_4(OH)_4(MAA)_{12}$ cluster) or Zr-MOCN, a composite membrane Zr-MOC@PP consisting of the PP separator as supporter (Supplementary Fig. 14) and mono-meric Zr-MOC was also prepared, in which the stretched pores of the PP was impregnated with Zr-MOC slurry. Voltage profiles

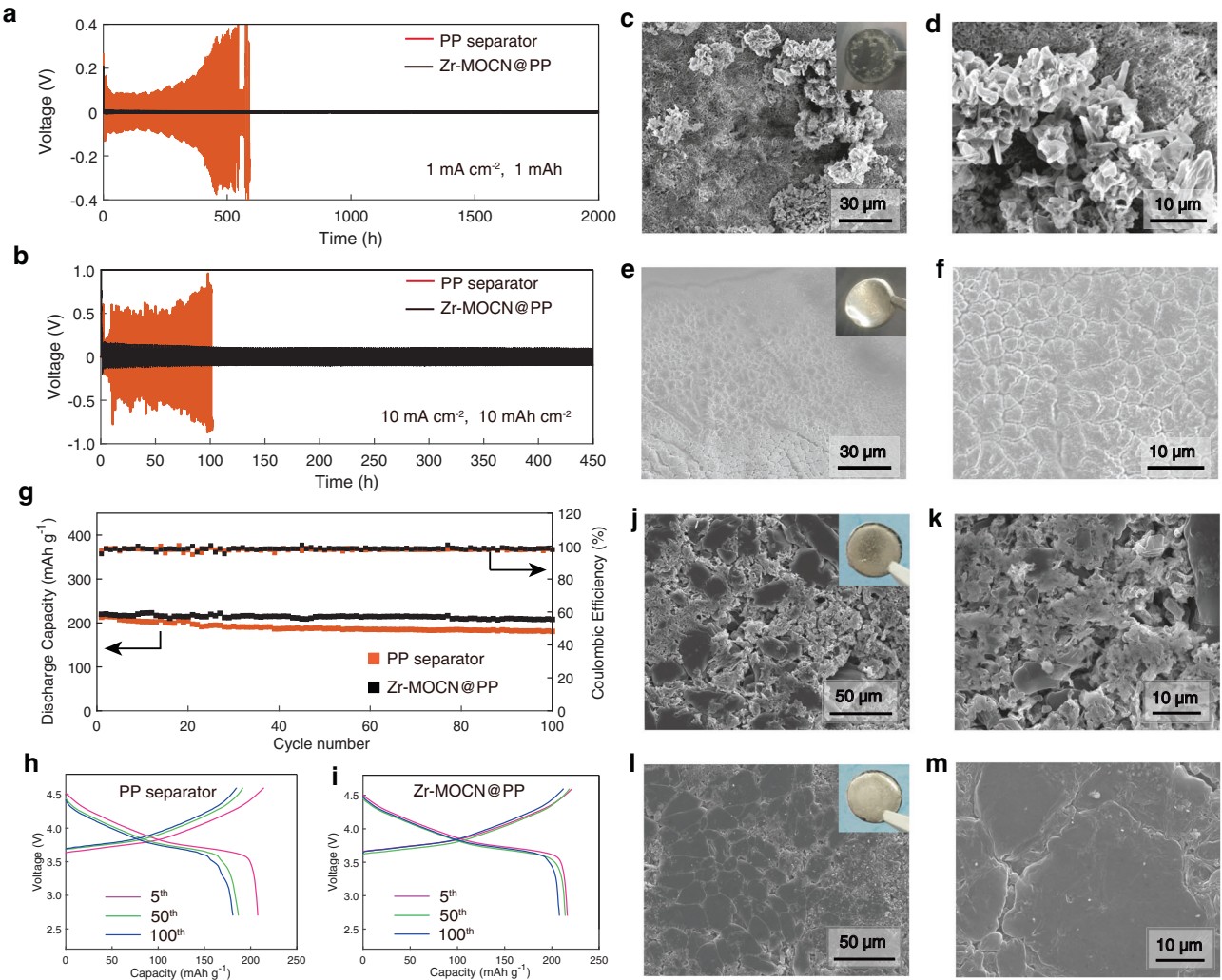

**Fig. 2 Electrochemical performance and morphology of cycled lithium metal in Li-symmetric cells and Li | NMC622 cells.** Initial voltage profiles of the PP separator (red) and Zr-MOCN@PP (black)-based cells at a fixed current density of: **a** 1 mA cm$^{-2}$. **b** 10 mA cm$^{-2}$. Top-view SEM images of the lithium metal after 340 h stripping/plating in **c**, **d**. The PP separator-based symmetric lithium cell; **e**, **f** Zr-MOCN@PP-based symmetric lithium cell; insets are photos of the corresponding lithium metal electrode. **g** Cycling stability and Coulombic efficiency of Li | NMC622 cells with the PP separator (red) and Zr-MOCN@PP (black) at 0.2 C rate; the voltage range of charge and discharge is 2.7–4.6 V. **h**, **i** Charge/discharge voltage profiles with the PP separator and Zr-MOCN@PP based cells, respectively. SEM images of Li metal after 100 cycles with: **j**, **k** PP separator, **l**, **m** Zr-MOCN@PP separator; insets are photos of the corresponding lithium metal electrode.

observed in the Li-symmetric cell using Zr-MOC@PP displayed a polarization slightly lower than that of the PP separator cell (Supplementary Fig. 15), however much higher than the Zr-MOCN@PP cell. The cell failed after 320 h cycling. Lithium dendrites could be easily observed even at the 120 h cycles under SEM, and a visible thick black layer was produced by the parasitic reactions (Supplementary Fig. 16). To rule out the possible effect of thickness and binders of Zr-MOC@PP, we added two additional control experiments using Zr-MOC immersed PP (detailed preparation methods in Supplementary Information). We found both Zr-MOC@PP-2 and Zr-MOC@PP-3 separators showed no obvious effect in suppressing lithium dendrite and parasitic reactions (Supplementary Figs. 17–19). Moreover, to directly observe the lithium deposition behavior, an in-situ optical microscopy measurement was performed with the pouch cell, which was constructed by thin lithium metal as anode, graphite as counter electrode, and a separator with excessive amount of LiPF$_6$-LE (Supplementary Fig. 20). Dendrites were immediately produced and crazily grew when using the PP separator (Supplementary Video 1; Supplementary Fig. 21c, d); while the

lithium was smoothly deposited on the Li surface in the Zr-MOCN@PP cell (Supplementary Video 2; Supplementary Fig. 21a, b), and no dendrite was observed. This result providing additional confirmation for the argument that Zr-MOCN@PP membrane effectively eliminated Li$^0$ dendrite formation.

Furthermore, the Li | Cu asymmetric cells with the PP separator and Zr-MOCN@PP were assembled to evaluate the CE of the Li plating/stripping in the carbonate-based liquid electrolyte (LiPF$_6$-LE). The cell with the Zr-MOCN@PP achieved much high CE (99.3%), which outperformed the performance with the PP separator (86.7%) (Supplementary Fig. 22).

The effectiveness of Zr-MOCN@PP was eventually subject to the test of an actual battery environment. A high voltage lithium-metal battery Li | NMC622 full cell using high-energy LiNi$_{0.6}$Mn$_{0.2}$Co$_{0.2}$O$_2$ cathode, Li metal anode, and LiPF$_6$-LE were constructed (Fig. 2g). The cell with the Zr-MOCN@PP separator was cycled at 0.2 C within the charge/discharge voltage range of 2.7–4.6 V. Impressively, after 100 cycles with the Zr-MOCN@PP separator the surface of Li maintained metallic lustre with no dendrite, in sharp contrast with the messy lithium surface

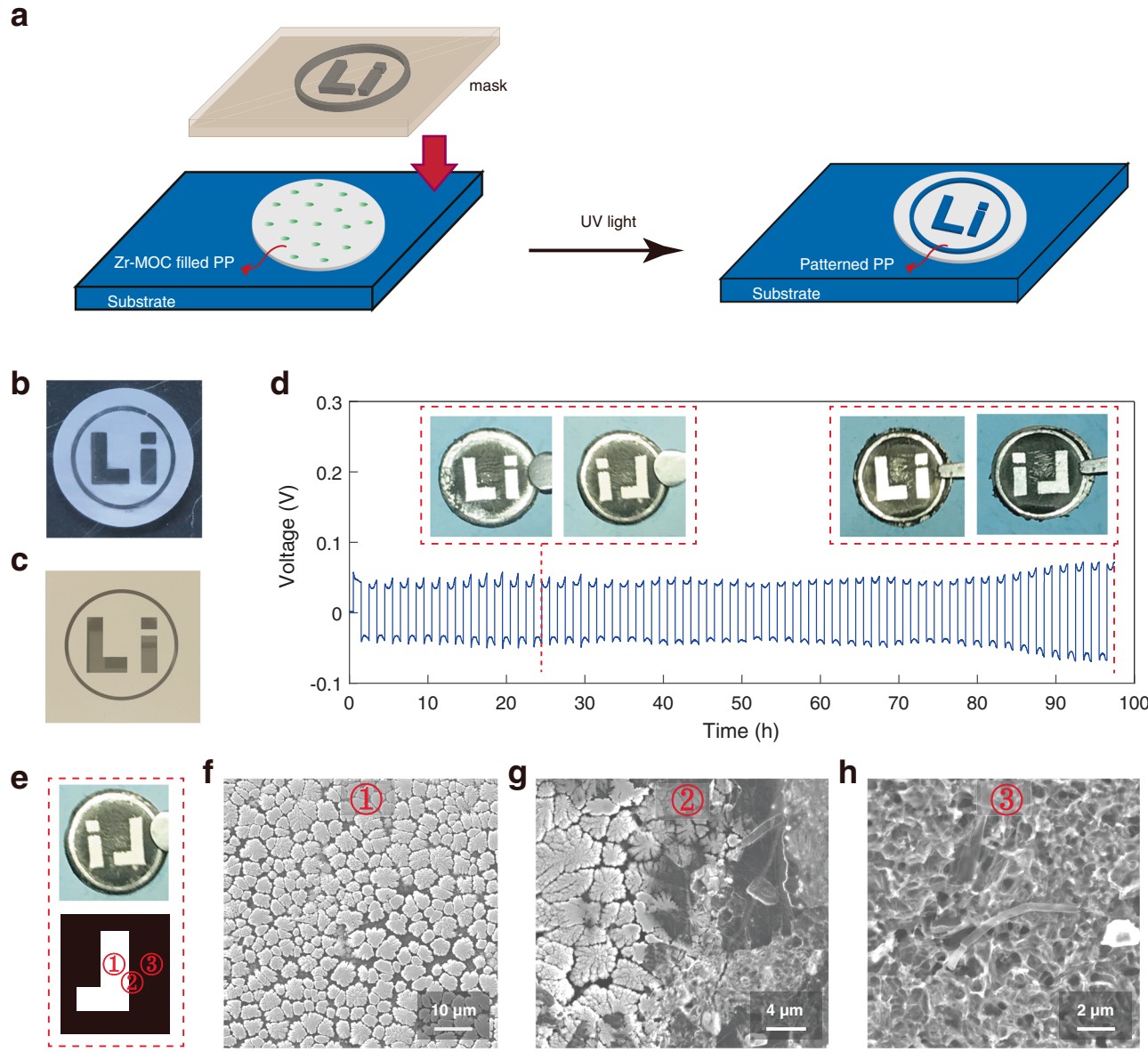

**Fig. 3 Preparation of patterned Zr-MOCN@PP separator and the electrodeposition properties. a** Schematic of the preparation process of the patterned separator. **b, c** The patterned separator and the photomask of "Li" shape, respectively. **d** Voltage vs. time profile of symmetric lithium cell with a "Li" shape pattern separator at a fixed current density of 1.0 mA cm$^{-2}$, each half-cycle lasts 1 h. Insets showed the lithium metal electrodes after 24 h and 96 h stripping/plating cycles, respectively. **e** Lithium metal electrode after being cycled for 24 h and scheme of 'L' with different positions of ①,②,③ in the lithium metal above. **f, g, h** Top-view SEM images of the lithium metal electrode of ①,②,③.

recovered from the cell that using PP separator (Fig. 2j-m). Electrochemically, the cell with the Zr-MOCN@PP membrane also significantly outperformed the PP separator (Fig. 2g-i), enabling a capacity retention of 94.6% after 100 cycles, which is much better than that PP-based cell (84.8%).

The effect of photoresist in deactivating Li$^0$-electrolyte reactivity is so strong that we can precisely control the formation of Zr-MOCN@PP film on the microscopic level. A "Li" shape pattern was transferred from a photomask (Fig. 3c) to the PP separator substrate using a contact lithographic technique that is mature in semiconductor manufacturing[27] (Fig. 3a). As shown in Fig. 3b, the pristine PP area is white, and the patterned area is transparent (against a black background). The patterned separator was then assembled into a symmetric lithium coin cell and cycled (Fig. 3d). The lithium metal electrode can be selectively protected with high spatial resolution by the patterned separator. Lithium surface

parasitic reactions were well suppressed at the Zr-MOCN patterned areas, and maintained the pristine metallic lustre after cycling, while the unprotected area stood out with the corroded surface (Fig. 3d). SEM images further revealed the detailed differences for the Zr-MOCN protected area ①, the pristine PP area ③, and their junction area ② (Fig. 3e). Lithium dendrite growth can be observed after 24 h striping/plating cycles in area ③, while at the junction area ②, these two different electrodeposition behaviors could be easily identified even at the micron scale (Fig. 3g). As the cycling times up to 96 h, lithium metal on the bare PP area became darker, indicating more parasitic reaction by-products accumulated, while the protected area remained shining metal lustre.

Chemical analyses were carried out on the protected and unprotected regions of the patterned lithium surface using time-of-flight secondary ion mass spectrometry (ToF-SIMS) and X-ray photoelectron spectra (XPS). In the depth profiles of ToF-SIMS

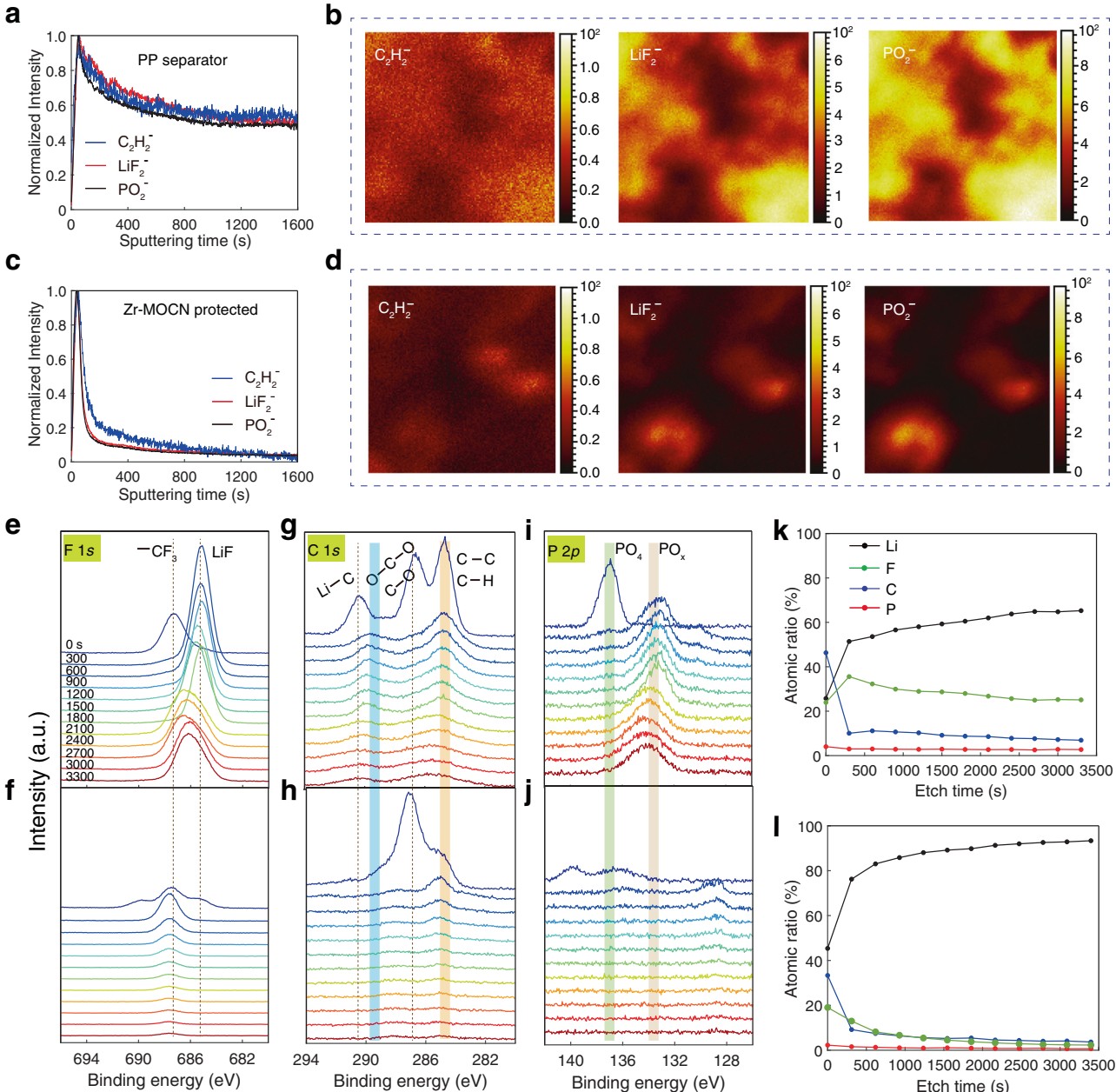

**Fig. 4 Characterization of chemical compositions on the cycled lithium metal electrodes by ToF-SIMS and XPS spectra. a**, **c** ToF-SIMS normalized (to maximum) depth profiling of several typical second ion fragments on the cycled lithium metal electrode surface of the pristine PP and Zr-MOCN protected areas, respectively. **b**, **d** ToF-SIMS chemical maps of second ion fragments with 1600 s of sputtering in the lithium metal electrodes cycled in the PP and Zr-MOCN protected areas, respectively. **e**-**j** XPS spectra of F, C and P elements recorded different time of argon ion sputtering at cycled lithium metal electrode surface of the pristine PP (upper panel) and Zr-MOCN (lower panel) protected areas, respectively. **k**, **l** The corresponding depth profiles of the atomic concentration of Li, F, C, and P elements.

(Fig. 4a, b; Supplementary Fig. 23), the pristine PP area ③ showed strong $C_2H_2^-$, $LiF_2^-$ and $PO_2^-$ signals, which arise from the reduction of electrolyte components, and remained 50% high even after the surface was sputtering by caesium ions for 1600 s (around 280 nm). While the signals of the Zr-MOCN protected area were quickly decreased within 150 s of sputtering (around 25 nm), the abundances of the corresponding electrolyte reaction products were lower than 20%. The 25 nm-thickness of the surface layer was ascribed to the reaction between the limited free electrolyte solvents (on the lithium metal surface) and $Li^0$. The number of inorganic species ($LiF_2^-$ and $PO_2^-$) and organic species ($C_2H_2^-$) on the surface of the cycled lithium foil of the

Zr-MOCN protected area were significantly lower than that of the bare PP area (Fig. 4c, d).

Chemical information from XPS depth profiles reveals that interphasial products (Fig. 4e-j) could be detected in the outermost layer of both pristine PP and Zr-MOCN protected areas. However, as the sputtering time exceeds 300 s (corresponding to ~23 nm), both organic and inorganic interphasial products essentially disappears at the Zr-MOCN protected area (lower panel of Fig. 4f, h, and j, and Fig. 4k, l), consistent with the results of ToF-SIMS. Clearly, the formation of Zr-MOCN induced certain unknown mechanism that completely alters the chemical natures of the involved species.

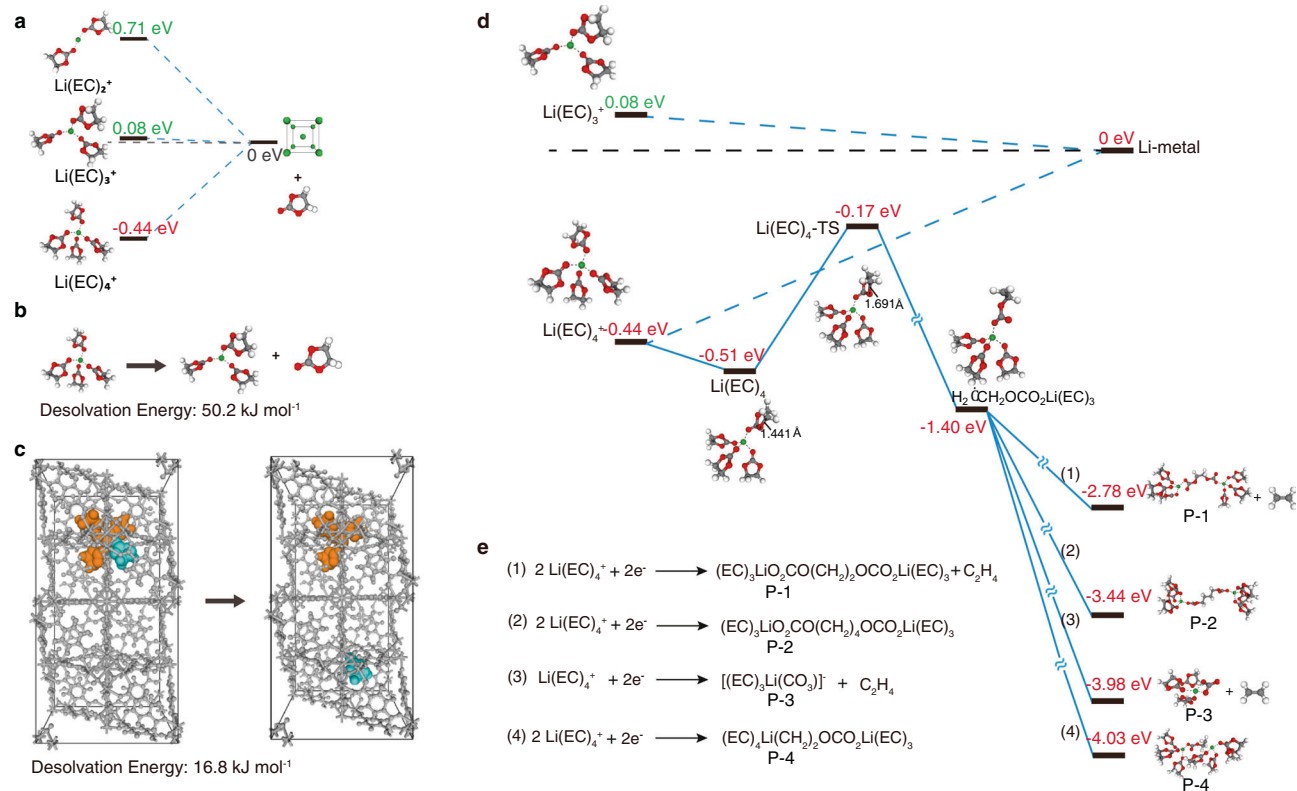

**Fig. 5 Density functional theory calculated energy profile for the reduction process of Li⁺-solvates. a** Reduction energy of Li⁺-solvates Li(EC)$_4^+$ and partially desolvated species Li(EC)$_3^+$, Li(EC)$_2^+$ by density functional theory (DFT). **b, c** Desolvation energy of Li(EC)$_4^+$ to Li(EC)$_3^+$ and EC molecule in liquid electrolyte and UiO-66, respectively. **d** The reduction process of Li(EC)$_4^+$ and Li(EC)$_3^+$ and reductive decomposition process of Li(EC)$_4^+$. **e** Typical Li(EC)$_4^+$ reductive decomposition reactions.

**The mechanism for parasitic reaction suppression.** Density functional theory (DFT) calculation was employed to understand the underneath mechanism of how the Zr-MOCN stabilizes Li$^0$ with electrolyte. As for the amorphous feature of the Zr-MOCN (Supplementary Fig. 24), studying the host-guest interaction for such transition-metal contained large scale system using current computational chemistry tools is quite challenging. Therefore, to simplify the porous structure of Zr-MOCN and save the computational cost, a highly symmetrical metal-organic framework (MOF) analogue, UiO-66 (Supplementary Fig. 25) was chosen as the host materials for modelling. Since the precursor of Zr-MOCN and UiO-66 consist of a Zr$_6$O$_4$(OH)$_4$ core and 12 carboxylate ligands, the structural similarity of Zr-MOCN and UiO-66 give them similar pore environment, which is one important factor for the host-guest interactions. The dissociation energy of Li(EC)$_4^+$ to Li(EC)$_3^+$ and one EC molecule on the UiO-66 substrate was about 16.8 kJ mol$^{-1}$, much lower than the corresponding dissociation energy in the bulk electrolyte environment (50.2 kJ mol$^{-1}$, Fig. 5b, c); suggesting that the Li$^+$-solvent complexes become partially dissociated or weakened in the presence of UiO-66 surface. The reduction potential of various solvate species including Li(EC)$_4^+$ and partially desolvated species Li(EC)$_3^+$, Li(EC)$_2^+$ were also calculated (Fig. 5a). Li(EC)$_4^+$, a commonly recognized primary Li$^+$-solvates in the electrolyte[28] was found to experience reduction at −0.44 V vs. Li$^+$/Li (Supplementary Table 6), indicating that around 42.5 kJ mol$^{-1}$ of electric energy was required to drive the reduction reaction. This estimated energy as well with the experimental values from the previous report[29]. However, the reduction potentials of the partially desolvated species Li(EC)$_3^+$ and Li(EC)$_2^+$ were calculated to be as low as 0.08 V and 0.71 V, respectively (Supplementary

Tables 7, 8). Hence, after one EC was dissociated, the reduction of the Li$^+$-solvate Li(EC)$_3^+$ to Li$^0$ became energetically competitive with the reduction of solvent molecules. Further dissociating EC would lead to a high energy state of Li$^+$-solvate species (Fig. 5a).

To further understand the correlation between dissociated Li$^+$-solvates and the parasitic reactivity, we also simulated the possible pathways of irreversible reactions. The primary parasitic reaction in the commonly used carbonate-based non-aqueous electrolyte is the reduction of EC in the primary Li$^+$-solvation sheath, which is the key reaction to form interphase that enables stable cycling of graphite, but at the same time problematic for Li$^0$. Several possible pathways for the decomposition reaction of the Li$^+$-solvates exist (Fig. 5e), involving one electron and probably two-electron reduction processes[19,30–32].

We calculated total energies and energy barriers for these possible reduction paths. As shown in Fig. 5d, the formation energy of products **P-1**, **P-2**, **P-3**, **P-4** was −2.78 eV, −3.44 eV, −3.98 eV, and −4.03 eV, respectively (Supplementary Tables 9–17); suggesting all these possible parasitic reactions could occur spontaneously from the thermodynamic perspective. However, all these decomposition reactions have to undergo an elementary reaction — homolytic C–O bond cleavage of EC (bond length stretched from 1.441 to 1.691 Å), which possesses an energy barrier of +0.34 eV. On the other hand, the reduction energy of Li(EC)$_4^+$ to Li metal and EC molecules was calculated to be 0.44 eV, higher than the energy barrier of the ring-opening reaction of EC. The solvent reduction reactions therefore were more likely to occur in the non-aqueous electrolytes. The nano-porous structure of UiO-66 or Zr-MOCN forced the partial desolvation of Li(EC)$_4^+$, likely via direct interaction between Li$^+$ and the OH-rich functionalities attached to the wall of Zr-MOCN. Such interaction becomes compelling due to the comparable sizes of

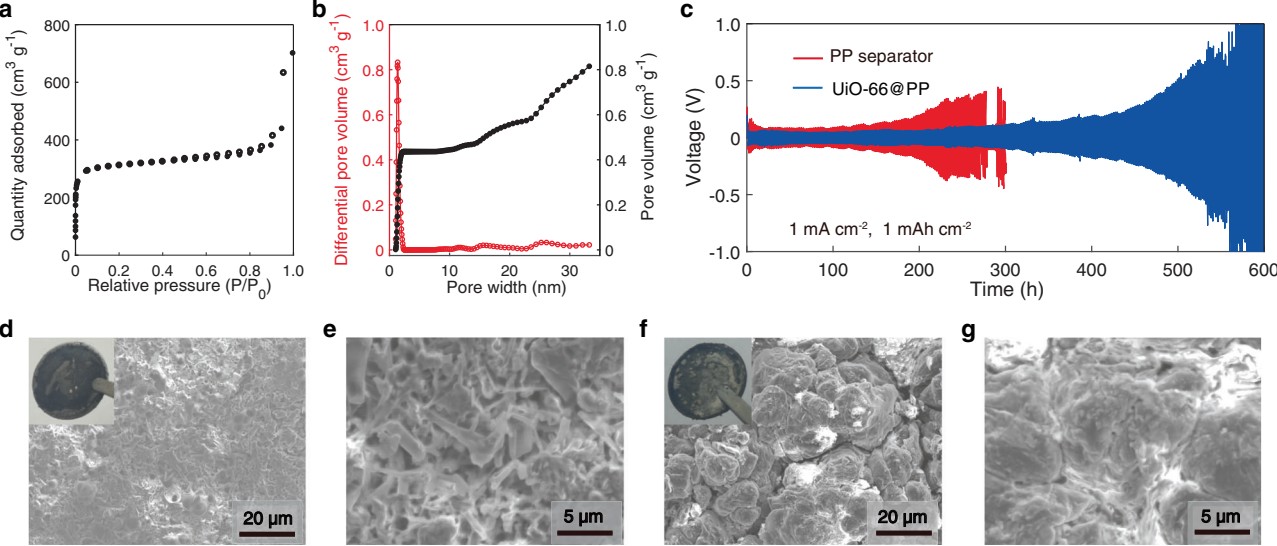

**Fig. 6 Electrochemical performance of UiO-66@PP separator. a, b** Nitrogen-sorption isotherm curves and pore size distribution (red) and cumulative pore volume (black) profiles measured at 77 K of UiO-66 after ball-milled, respectively. **c** Voltage vs. time for symmetric lithium cell where each half-cycle lasts 1 h. Initial voltage profiles of the PP separator (red curve), UiO-66@PP (blue curve)-based cells at a fixed current density of 1 mA cm$^{-2}$ (1 mAh cm$^{-2}$). SEM images of the lithium metal anode after 120 h cycling: **d, e** The PP separator-based symmetric lithium cell. **f, g** UiO-66@PP-based symmetric lithium cell. The insets show the lithium metal after 120 h cycling in different cells: the PP separator and UiO-66@PP-based symmetric lithium cells, respectively.

the Zr-MOCN nanopores and the primary solvation sheath of Li$^+$. Energetically, the resulted partially desolvated species would favor the reduction of Li$^+$ rather than the solvent molecules solvating Li$^+$, thus circumventing the excessive solvent reduction and their corresponding footprint in interphases.

Therefore, a question arises: could the UiO-66 separator suppress the lithium dendrite and continuous parasitic reaction effectively? We synthesized UiO-66 in the form of powder, which has a BET surface area of 1240 m$^2$ g$^{-1}$ and a pore size of 0.7–0.9 nm (Supplementary Fig. 26). To use this MOF powder as a free-standing separator, UiO-66 was ball-milled and fabricated as slurry to impregnate the pores of the PP separator (denoted as UiO-66@PP). The cell with the UiO-66@PP membrane (Supplementary Fig. 27) exhibited much lower voltage polarization (around 60–100 mV) than that of the PP separator based cell (around 240–400 mV) (Fig. 6c). The post-mortem SEM analysis of the cycled lithium electrode showed the lithium dendrite growth was significantly alleviated in the presence of UiO-66-containing PP (Fig. 6d-g). The sign of Li$^+$-electrolyte reactivity was only visible on a fraction of the Li-metal surface; suggesting the parasitic reactions might be partially inhibited. The still existing but significantly reduced Li$^0$ corrosion could be ascribed to the re-saturated low-energy state solvation species due to the existence of gaps between UiO-66 particles. Although the UiO-66 powder was thoroughly ball-milled, the gaps were still inevitable. The QSDFT pore size distribution[33,34] revealed the existence of large pores with a size ranging from 10 to 50 nm and a pore volume of 0.7 cm$^3$ g$^{-1}$, almost two times higher than the contribution of inner pores of UiO-66 (Fig. 6a, b).

From Raman spectroscopy (Supplementary Fig. 28), EC molecules in the bulk liquid electrolyte (Supplementary Fig. 28d) can be characterized by peaks at vibrational wavenumbers around 892 and 904 cm$^{-1}$, which are assigned to stretching vibration of C–O bond in free EC and Li$^+$-solvated EC (Li(EC)$_4^+$), respectively[35,36]. In the electrolyte-filled UiO-66, the peak ascribed to the Li$^+$-solvated EC was shifted from 904 to 898 cm$^{-1}$, corresponding to the partial dissociation of Li$^+$-EC solvates (Supplementary Fig. 28b). However, a peak at 904 cm$^{-1}$ did not disappear, suggesting a considerable amount of Li(EC)$_4^+$ still remain in the gaps between the UiO-66

particles. Zhou et al. recently reported that the MOF-modified electrolyte has a high Li$^+$ transference number, and the MOF-coated cathode has a CEI-free surface after long cycling[26,37,38], both of which benefited from partially desolvated lithium ions. It worth noting that the transference number ($t_{Li+}$) of the Zr-MOCN@PP cell in this work was also measured to be 0.69 (Supplementary Fig. 29), which is higher than the conventional liquid electrolyte system (~0.2–0.4)[39,40]. While these nano-sized pores in MOF/COF materials promote the desolvation process of Li$^+$-solvates in the liquid electrolyte[41], the large gaps between the MOF/COF particles would still allow the existence of fully solvated species, making parasitic reaction inevitable.

For the MOF crystallites (UiO-66) coated separator, the gaps existed not only between the separator and Li-metal surface, but also inside the UiO-66@PP due to its crystal feature (Fig. 7a; Supplementary Fig. 27). And these gaps were filled with free liquid electrolyte. Thus, the Li$^0$-electrode section, which is closely contacted (at the molecular level) with porous materials, its Li$^0$-electrodeposition behavior will be tuned. But the gaps surrounding section, partially desolvated Li$^+$-species will be re-saturated, resulting in the continuous occurrence of parasitic reactions and generation of by-products (Fig. 6f; Fig. 7a). In contrast, the surface of Zr-MOCN@PP was remarkably smooth (Fig. 1f, j, k, with an R$_q$ of 1.58 nm and R$_{max}$ of 12.5 nm), due to the high-resolution photoresist feature of its precursor (Zr-MOC, Supplementary Figs. 3–5). Further, the roughness majorly came from the Zr-MOC filled substrate (PP separator). If Zr-MOC was coated on a flat substrate, such as a silicon wafer, the resulting film could be extremely smooth (Supplementary Fig. 3).

This feature leads to very few gaps existing inside the Zr-MOCN@PP separator (Fig. 1g). As for the surface contact gaps between Zr-MOCN@PP and Li$^0$-electrode, they still exist due to the inevitable surface roughness of both (Fig. 7b); but these gaps (and the free electrolyte in the gap) are significantly fewer than the case of UiO-66@PP separator. In the closely contacted section (of Zr-MOCN@PP and Li$^0$-electrode), the partially desolvated Li$^+$-species are more easily reduced to Li-metal, and thus deposited preferentially, which could have a repair effect to surrounding gaps. Therefore, in the Zr-MOCN@PP based cells,

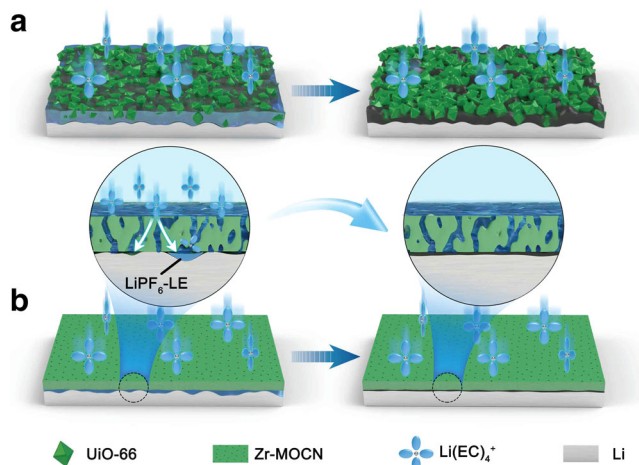

**Fig. 7 Schematic illustration of the Li-deposition in cells with different separators. a** The UiO-66@PP. **b** Zr-MOCN@PP.

the parasitic reaction cannot be completely ruled out, as we can find signs of by-products from XPS and ToF-SIMS; but their continuous occurring was effectively suppressed (Fig. 2e, f; Fig. 4).

Finally, to verify the universality of this nano-confinement approach, Zr-MOCN@PP membrane is tested with ether-based electrolyte (Supplementary Fig. 30a), which has been known to be less reactive with Li$^+$ than carbonates. Although the lower reactivity of ether solvents indeed results in a less-dendrite Li morphology even in the absence of Zr-MOCN@PP, a blackened surface was still observed on the recovered Li$^0$ after 200 h cycling (Supplementary Fig. 30b). Again, in sharp contrast, the metallic lustre remained on the Li$^0$ recovered from the cell containing Zr-MOCN@PP membrane, with no dendritic Li$^0$ as revealed by the SEM image (Supplementary Fig. 30c).

We designed a uniform nano-porous membrane via in-situ photopolymerization of Zr-MOC, whose nano-structured pores was found to force partial desolvation of Li$^+$ and subsequently alters the reactivity between Li$^0$ and carbonate or ether electrolytes. Such suppressed reactivity enables a highly reversible Li$^0$ electrode, as evidenced by both symmetric lithium cell as well as actual lithium metal battery based on a high nickel cathode, which can be cycled for thousands of hours without dendritic or dead Li$^0$ formation as well as excessive solvent reduction. The high spatial resolution of the photoresist even allows for precise patterning on Li$^0$ surface, thus making it possible to print lithium-metal batteries at nano-scale. The direct suppression of Li$^0$ reactivity with electrolytes via nano-confinement provides an alternative but highly effective pathway toward the high energy density battery.

## Methods

**Synthesize of Zr-MOC**. 70 wt% Zr(OPr)$_4$/1-propanol solution (20.0 g) and MAA (20.0 g) were mixed in a 150 mL flask, stirred at room temperature for 5 min. Then stirring was stopped, the solution was heated at 80 °C for 18 h. After that, the temperature was lowered to 60 °C and kept for 6 h, 40 °C for 6 h and room temperature for 2 days. A large amount of colorless crystals generated, which were isolated from solution and dried in a vacuum oven at room temperature for 6 h, the yield was 87%.

**Synthesize of UiO-66**. A Zirconium tetrachloride (ZrCl$_4$) (0.6 mmol, 0.1398 g) and 1,4-benzenedicarboxylic acid (0.6 mmol, 0.0997 g) dissolving in acetic acid (27.3 g, 455 mmol) and N, N-dimethylformamide (DMF) (24.9 g, 340 mmol) at room temperature. The obtained mixture was sealed and heated at 120 °C for 20 h. The white solid precipitated from the solution, washed with DMF several times and methanol, then dried at room temperature.

**Preparation of Zr-MOCN@PP porous membrane**. The prepared Zr-MOC was dissolved in PGMEA to give a 50 wt/wt% solution, then 2,4,6-trimethyl benzoyl-diphenyl phosphine oxide (TPO) (1 wt% to the Zr-MOC) was added in the above

solution. After that, the PP separator was immersed in the mixture, and the soaked PP separator was taken out, exposed to 365 nm mid-UV radiation for 30 min under the Argon atmosphere, which resulted in the formation of a transparent membrane. Finally, the membrane samples were soaked in LiPF$_6$-liquid electrolyte for 24 h.

**Preparation of Zr-MOC@PP separator**. The composite Zr-MOC@PP membrane was fabricated by a slurry-coating method. The Zr-MOC and PVDF were dispersed in NMP to make a slurry with a ratio of 9:1 (by weight) at room temperature. Then the slurry was stirred 1.0 h and coated onto one side of the PP separator with a doctor blade. The slurry-coated separator was dried in an oven at 60 °C for 10 h

**Preparation of UiO-66@PP separator**. The UiO-66@PP composite separator was fabricated by a slurry-coating method, which was similar to that of Zr-MOC@PP separator. The UiO-66 and PVDF were dispersed in NMP to make a slurry with a ratio of 9:1 (by weight) at room temperature. Then the slurry was stirred 1.0 h and coated onto one side of the PP separator with a doctor blade. The slurry-coated separator was dried in an oven at 60 °C for 10 h. Before preparing the slurry, the UiO-66 powder was ball-milled to obtain smaller crystallite, which could be useful to form a more uniform separator.

**Materials characterizations**. Nitrogen-sorption isotherms were measured at 77 K with a Quantachrome IQ2 Instrument Corporation model 3Flex surface characterization analyzer. Single X-ray diffraction data was recorded on a Bruker P4. The microscopic morphologies of the samples were characterized by field emission scanning electron microscopy (ZEISS Gemini, 5 kV, Germany). The AFM images were taken by Bruker Icon and the data were analyzed by NanoScope Analysis 1.8. Optical microscopy images were taken by optical microscope of Axio Lab A1 (ZEISS, Germany). Electron-Beam (e-beam) exposure was performed by NB-5 electron-beam lithography system in Tsinghua nanofabrication technology center. The Raman spectra were measured by using a Raman spectrometer (JY LabRam HR-800, Horiba Jobin Yvon, France). Time-of-flight secondary ion mass spectrometry (ToF-SIMS) was applied to characterize the by-products formed on the cycled lithium metal electrode via ToF-SIMS5 (ION-ToF-GmbH, Germany). A pulsed 30 KeV Bi$^+$ ion beam was set, and the selected analysis area was 100 × 100 μm. Then the 500 eV Cs$^+$ ion beam with the incident angle of 45° (sputtering rate was 0.173 nm s$^{-1}$ for SiO$_2$) was applied to sputtering the cycled electrodes. Before measurement, all of the samples were lightly rinsed with DMC to remove any trace amount of salt on the electrode surface and dried in an argon-filled glove box. Then the samples were carefully pasted on a ToF-SIMS-holder in the glove box and stored it in an argon-filled box, rapidly transferred the holder from the box to the ToF-SIMS ultra-high vacuum analysis chamber. X-ray photoelectron spectroscopy (XPS) measurements were performed on 250XI using Al Ka radiation (72 W, 12 kV) at a pressure of 10$^{-9}$ torr and an argon ion beam. Depth profiling was fulfilled using Ar ion sputtering at the rate of 4.5 nm per minute for Si. Powder X-ray diffraction data was performed on a D8 Advance Brooker Ultima III diffractometer from 2θ = 4.0° up to 60° with a step size of 0.02° and 0.5 s per step.

**Electrochemical measurements**. The galvanostatic charge/discharge tests were performed using CT2001A cell test instrument (LAND Electronic Co. Ltd) at room temperature for the liquid electrolyte cells. The current density of Li|Li symmetric cells with liquid electrolyte is 1.0 mA cm$^{-2}$, and 10 mA cm$^{-2}$, respectively. All the batteries were assembled in an Argon-filled glove box. The in-situ optical microscopy measurement was performed with the pouch cell, by using lithium metal and graphite as electrodes. The 1.0 M LiPF$_6$ in EC/DMC/EMC (by weight) and 1 wt% of VC was used as liquid electrolyte for the consistency of the experiment.

The pouch cell was fully charged before observation, and then it was discharged at a fixed current density of 1.0 mA cm$^{-2}$ during measurement. The cathodes were made up of LiNi$_{0.6}$Mn$_{0.2}$Co$_{0.2}$O$_2$ (NMC622) active material, super P and PVDF with a mass ratio of 8:1:1, and NMP was applied as the dispersant. The active materials' weight density for a NMC622 electrode sheet is 4.5 mg cm$^{-2}$, and the areal capacity loading is 0.855 mAh cm$^{-2}$. 1.0 M LiPF$_6$ in EC/DMC/EMC (1/1/1, w/w/w) and 1 wt% of VC was used as liquid electrolyte. The NMC622 with the diameter of 12 mm was used as a cathode, the lithium metal foil with the thickness around 400 μm and the diameter of 15.4 mm was used as an anode (used as received without any treatment). The pressure was 50 kg cm$^{-2}$ for cell stack. The voltage range of charge and discharge is 2.7–4.6 V. All the cells were conducted for formation cycles at 0.1 C for two cycles followed by 0.2 C for cycling.

**In-situ optical microscopy measurements**. Firstly, the pouch cell was constructed by thin lithium metal as anode, graphite as counter electrode and Zr-MOCN@PP or PP as a separator (Supplementary Fig. 20). Then, the cell was clamped and cut carefully to obtain a smooth section surface. After that, a large amount of liquid electrolyte (LiPF$_6$-LE) was filled through the cross-section. Finally, the sample was assembled into the testing system for observation. It should be noted that several short circuit tests were required during the sample preparation to ensure the cell could charge/discharge well. The graphite anode was charged before observation;

followed that the cell discharged at the current density of 1.0 mA cm$^{-2}$ to observe the lithium electrodeposition behavior.

**Theoretical calculation methods**. Crystalline structure of UiO-66[42] was generated using single crystallographic data from the Cambridge Structural Database (CSD entry: 733458). The structure of pristine UiO-66, guest molecules accommodated UiO-66 (Li(EC)$_4^+$@UiO-66, Li(EC)$_3^+$@UiO-66, EC@UiO-66) were first calculated using the DFT[43,44] implemented in the CASTEP[45] module of Materials Studio. The generalized gradient approximation in the form of Perdew–Burke–Ernzerhof (PBE)[46] was selected as the exchange-correlation functional. Grimme dispersion correction[47,48] was employed in all calculations to describe van der Waals (vdW) interactions. A plane wave energy cutoff of 830 eV and the Monkhorst-Pack k-point grid of $1 \times 1 \times 1$ were used. To include the contribution of the electrolyte environment, the CASTEP optimized structures were then calculated with implicit solvent model using DFT-COSMO (DFT based conductor-like screening model) method[49,50]. DFT-COSMO calculations were performed using DMol3 module[51] of Materials Studio. Double Numerical basis with Polarization functions (DNP) was selected as the basis set; GGA-PBE[46] was selected as the exchange-correlation functional. Grimme dispersion correction[47,48] was employed in all calculations to describe vdW interactions. The COSMO implicit solvation model with acetone parameters (with a dielectric constant, ε = 20.7) was used to represent a typical mixed solvent electrolyte environment. The desolvation energy ($E_{desol}$) of Li(EC)$_4^+$ in the pores of UiO-66 was calculated as $E_{desol} = E_{[Li(EC)_{3+}@UiO-66]} + E_{[EC@UiO-66]} - E_{[Li(EC)_{4+}@UiO-66]} - E_{[UiO-66]}$, where the $E_{[UiO-66]}$, the corresponding DFT calculation model was shown in Scheme 1c. To investigate the reduction process of Li$^+$ and possible parasitic reactions, we used the combination of explicit and implicit solvents model to reflect the actual solvation environment. This model includes the first solvation shell of Li$^+$; while the effect of the solvent beyond the first solvation was represented by the SMD solvation model, implemented using Gaussian 16 package[52]. 6-31 + G (d,p) was selected as the basis set; PBE was selected as the exchange-correlation functional for DFT calculations. The PBE functional chosen here is because of its more accurate predictions of the Li$^+$-solvent binding energy than the popular DFT functionals, including B3LYP and M05-2X[53]. Density functional correction (DFT-D3 with Becke-Johnson damping, DFT-D3(BJ))[54] was employed in all calculations to describe vdW interactions. Acetone parameters were used to represent the commonly used electrolyte in the SMD models[55]. All simulation works were performed using the computing resources at the National Supercomputing Center in Shenzhen.

## Data availability

All data are available in the paper or the supplementary materials from the corresponding author upon reasonable request.

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

## Acknowledgements
Research at Tsinghua University was funded by the National Natural Science Foundation of China (52073161, H.X. and U1564205, X.H.) and the Ministry of Science and Technology of China (No. 2019YFE0100200, X.H. and 2019YFA0705703, L.W.). Research at Argonne National Laboratory was funded by the US Department of Energy (DOE) Vehicle Technologies Office under Contract No. DEAC02-06CH11357, G.-L.X. and K.A. The authors thank Joint Work Plan for Research Projects under the Clean Vehicles Consortium at U.S. and China–Clean Energy Research Center (CERC-CVC2.0, 2016-2020), and thank Tsinghua University-Zhangjiagang Joint Institute for Hydrogen Energy and Lithium Ion Battery Technology. The authors also thank Yinglong Chen (Tianjin LISHEN battery Joint-stock Co., Ltd), who helped to do the in-situ optical microscopy measurements.

## Author contributions
L.S. performed materials synthesis and characterization, electrochemical testing; Q.W. and X.W. fabricated the photomask and tested photoresist performance; H.C. carried out computational calculations; Y.X. and Z.L. performed UiO-66 synthesis and BET measurement. L.S., X.L., L.W., Z.C., G.-L.X., and X.H. contributed to the electrochemical analysis and discussion; L.S., J.W., Y.T., K.A., H.X., and X.H. contributed to interpreting the parasitic reaction mechanism. L.S., H.X., and X.H. wrote the paper. The project was directed and supervised by H.X., and X.H. All authors contributed to discuss the results.

## Competing interests
The authors declare no competing interests.
