## [Peer Review File · Nature Communications]

REVIEWER COMMENTS

Reviewer #1 (Remarks to the Author):

0) The authors present a very compelling separator technology. The work is significant and warrants publication in Nature Communication provided the concerns detailed below are addressed.

1) Using Raman the authors come to the conclusion that fully solvated $\text{Li}(\text{EC})_4^+$ remains in the gap between UiO-66 particles (line 293) and this is the reason for worse performance. Supplementary Figure 9 shows pristine Li metal surface with surface roughness features larger than the gaps between UiO-66 particles. Please explain how the gaps formed between the rough Li metal surface and the Zr-MOCN separator do not allow for the presence of fully solvated Li^+ ? How close does the Li metal surface have to be to the Zr-MOCN for the author's proposed mechanism to work?

2) The videos are excellent, however, please provide details on what we're seeing and how the experiment was conducted. Explaining the cell structure with a schematic would be helpful.

3) P-Zr-MOC has pore size between 1.41 – 2.77nm. UiO-66 has pore size of 0.7 – 0.9nm, which is smaller than a fully solvated Li^+ as cited by the authors (line 84). Please explain in more detail why modeling efforts with UiO-66 are applicable to Zr-MOCN. How does pore size affect surface interaction and how does fully solvated Li^+ fit into smaller UiO-66 pore?

4) How does VC additive factor into all of this? Won't it preferentially decompose on Li regardless?

5) (line 460) Please provide areal capacity (mAh/cm^2) of the cathode so that mA/cm^2 can be calculated using reported c-rates. This information would be helpful to understand why Li deposits are an order of magnitude larger in the full cell than in the symmetric cells (Figure 2 f versus m). Information such as electrolyte loading, n/p ratio, and cell stack pressure is also critical information to include.

6) (Figure 2) It's surprising that filling the Celgard pores with a MOCN reduces the resistance of the separator. EIS testing of the symmetric cells would be useful to differentiate between R_{ct} of the Li electrode and the Ohmic resistance of the separator. The authors

Reviewer #2 (Remarks to the Author):

The authors have reported that the direct suppressing electrolyte and lithium metal reactivity by producing nano-structured photoresist membrane of porous Zr-MOC. They found out the polymerized Zr-MOC membrane was enough to stable and dense for protecting Li metal surface from Li dendrite formation with DFT calculation of the suppressing mechanism. It is interesting to study the inhibition of dendrite formation in Lithium-metal batteries by synthesizing nano-porous separators using photoresist. However, the structural analysis of Zr-MOC and Zr-MOCN was insufficient, and the flow of the overall paper was cluttered. It is interesting to calculate the mechanism that inhibits dendrite formation, but it is questionable whether the UiO-66 as a model is appropriate. It is recommended to reject this paper in current form and the following items are detailed comments.

1. The labeling of the research product was insufficient throughout the thesis. It may reduce the reader's understanding of this article. For example, lines 82-86 on page 4 describe the BET surface area. The first indication was that the polymerized Zr-MOC was p-Zr-MOC, but the relevant supplementary data was Zr-MOCN. In addition, the notation of Zr-MOCN began in the next paragraph. Therefore, it is recommended to clearly denote the product.
2. This is the same as point 1. Please clarify the structural explanations for $Zr_6O_4(OH)_4(MMA)_{12}$ cluster or p-ZrMOC on page 6 in line 138.
3. The synthesized Zr-MOC was described as a crystal structure, but there was no evidence to support it, such as an x-ray diffraction pattern. The CCDC number mentioned was insufficient and supplemental data are required.
4. In Figures 2a and b, there were no explicit polarization values are mentioned as in Figure 6c (eg. 60-100 mV, 240-400 mV). It would be helpful to compare each data by complementing exact values.
5. In Figure 4b and d, it is recommended to add the scale of the ToF-SIMS data.
6. The mechanism for parasitic reaction suppression was calculated with DFT method with UiO-66. However, in Supplementary Figure 15, UiO-66 and Zr-MOCN had different structure. The UiO-66 is a particle with continuously arranged pores, but the Zr-MOCN described in this text is a continuously

connected membrane by polymerization of Zr-MOC. I recommend to explain of the point at which Zr-MOCN can be structurally simplified to UiO-66.

7. Also, there is no detailed description of what arrangement or structure Zr-MOCN actually has in the text.

8. The preparation of Zr-MOCN membrane and Zr-MOC@PP separator were very different. One used immersion method and the other used doctor blade method. It might have different thickness of Zr-MOC and the use of different binders can lead to different electrochemical reactions. It is recommended to compare the results using a synthetic method similar to Zr-MOCN membrane.

Reviewer #3 (Remarks to the Author):

Authors designed a uniform nanoporous separator via in-situ photopolymerization of ZrMOC for the development of Li metal batteries. The Li symmetric cells with separator exhibit the better performance at a current density of 10mA cm^{-2} in the conventional carbonate-based electrolytes. The nano-structured pores in the separator was found to force partial desolvation of Li^+ and alter the reactivity between Li metal and electrolyte. However, some of the descriptions/experiments are questionable. Authors are encouraged to provide more experimental evidence and physical/chemical insights in the proposed mechanism.

1. Authors used excess Li (thick Li metal) which exceeds the capacity of NMC cathode in the Li metal batteries. Also, the vast Li sources leads to inflated Coulombic efficiency of the Li symmetric cells and decreases the energy density of the full cells. Thus, authors should examine the Li-Cu/Li-SS asymmetric cell to prove the significance of nanoporous separator in the carbonate-based electrolyte.

2. In the fundamental understanding, Li dendrites are formed at the Sand's time when the current density is over limiting current density. The formation of Li dendrites can be suppressed under low current density. Also, high Li^+ transference number (t_{Li^+}) plays crucial roles in the Li plating-stripping process. However, authors only mentioned that MOF-modified separator exhibits higher Li^+

transference number in the manuscript and did not provide the detailed discussion/experimental evidence regarding the limiting current/ Li⁺ transference number.

3. Nanoporous separator has been used to suppressed the formation of Li dendrites and stabilize the Li plating-stripping process. Authors did not compare their results with other studies. The literature review is not completed in this work.

4. The electrochemical impedance spectroscopy results of separators (PP and nanoporous separators) should be discussed during cycling.

Point-to-point Response to Reviewers

Reviewer 1:

Q0: The authors present a very compelling separator technology. The work is significant and warrants publication in Nature Communication provided the concerns detailed below are addressed.

We appreciate the reviewer's recognition of our work and helpful suggestions.

Q1: Using Raman the authors come to the conclusion that fully solvated $\text{Li}(\text{EC})_4^+$ remains in the gap between UiO-66 particles (line 293) and this is the reason for worse performance. Supplementary Figure 9 shows pristine Li metal surface with surface roughness features larger than the gaps between UiO-66 particles. Please explain how the gaps formed between the rough Li metal surface and the Zr-MOCN separator do not allow for the presence of fully solvated Li^+ ? How close does the Li metal surface have to be to the Zr-MOCN for the author's proposed mechanism to work?

We appreciate the reviewer's suggestions, which helped us improve the quality of our work. Both UiO-66 and Zr-MOCN are porous materials, but they behave very differently when used as separators for Li^+ -electrodeposition. It's a very good question; to answer it and also make the manuscript easy to understand for readers, we added a scheme (Fig. 7) in the revised manuscript. As shown in Fig. 5, the fully solvated Li^+ -solvates (for example, $\text{Li}(\text{EC})_4^+$) will be partially desolvated in the porous material, and show a higher tendency to be reduced. Thus, the Li-electrodeposition, which is closely contacted (at the molecular level) with porous materials, its Li^+ -electrodeposition behaviour will be tuned. But if there is a gap between the porous material and lithium electrode, the partially desolvated Li^+ -species will be fully solvated again during the free electrolyte in the gap.

Therefore, for the cells using MOF crystallites (UiO-66) coated separator, we found the parasitic reactions were alleviated, compared to pristine PP separator based cells (Fig. 6d-g). By-products covered only a portion of the Li-metal surface; the existing by-products could be ascribed to the re-saturated low-energy state solvation species due to the existence of gaps between UiO-66 particles. As we mentioned in the manuscript, although the UiO-66 powder was thoroughly ball-milled, the gaps were still inevitable (Supplementary Fig. 27) and N_2 -sorption measurement (Fig. 6a,b)). It is worth noting that, the gaps existed not only between the UiO-66@PP separator and Li-

electrode, but also inside the UiO-66@PP (Fig. 7a and Supplementary Fig. 27). Therefore, the parasitic reactions could continue to occur during the cell cycling, and eventually, produce a large number of by-products (Fig. 7a and Fig. 6).

In contrast, the surface of Zr-MOCN@PP was remarkably smooth (Fig. 1f,j,k, with an R_q of 1.58 nm and R_{max} of 12.5 nm), due to the high-resolution photoresist feature of its precursor (Zr-MOC, Supplementary Fig. 3-5). Further, the roughness majorly came from the Zr-MOC filled substrate (PP separator). If Zr-MOC was coated on a flat substrate, such as a silicon wafer, the resulting film could be extremely smooth (Supplementary Fig. 3). This feature leads to very few gaps existing inside the Zr-MOCN@PP separator (Fig. 1g). As for the surface contact gaps between Zr-MOCN@PP and Li-electrode, they still exist due to the inevitable surface roughness of both (Fig. 7b); but these gaps (and the free electrolyte in the gap) are significantly fewer than the case of UiO-66@PP separator. In the closely contacted section (of Zr-MOCN@PP and Li-electrode), the partially desolvated Li^+ -species are more easily reduced to Li-metal, and thus deposited preferentially, which could have a repair effect to surrounding gaps. And the cycled Li-electrode using Zr-MOCN@PP, exhibited a smooth surface (Supplementary Fig. 13d). It should be pointed out that the desolvated Li^+ -species will be re-saturated in the free liquid electrolyte, and the presence of gaps and accompanied free electrolyte between Li-electrode and Zr-MOCN@PP is inevitable. Therefore, in the Zr-MOCN@PP based cells, the parasitic reaction cannot be completely ruled out, as we can find signs of by-products from XPS and ToF-SIMS; but their continuous occurrence was effectively suppressed (Fig. 2e,f; Fig. 4)

Fig. 7 Schematic illustration of the Li-deposition in cells with different separators: **a**, the UiO-66@PP. **b**, Zr-MOCN@PP.

Accordingly, additional discussion on the mechanism was added in the revised manuscript on

Line 334-354:

“For the MOF crystallites (UiO-66) coated separator, the gaps existed not only between the separator and Li-metal surface, but also inside the UiO-66@PP due to its crystal feature (Fig. 7a; Supplementary Fig. 27). And these gaps were filled with free liquid electrolyte. Thus, the Li⁰-electrode section, which is closely contacted (at the molecular level) with porous materials, its Li⁰-electrodeposition behaviour will be tuned. But the gaps surrounding section, partially desolvated Li⁺-species will be re-saturated, resulting in the continuous occurrence of parasitic reactions and generation of by-products (Fig. 6f; Fig. 7a). In contrast, the surface of Zr-MOCN@PP was remarkably smooth (Fig. 1f,j,k, with an R_q of 1.58 nm and R_{max} of 12.5 nm), due to the high-resolution photoresist feature of its precursor (Zr-MOC, Supplementary Fig. 3-5). Further, the roughness majorly came from the Zr-MOC filled substrate (PP separator). If Zr-MOC was coated on a flat substrate, such as a silicon wafer, the resulting film could be extremely smooth (Supplementary Fig. 3).

This feature leads to very few gaps existing inside the Zr-MOCN@PP separator (Fig. 1g). As for the surface contact gaps between Zr-MOCN@PP and Li⁰-electrode, they still exist due to the inevitable surface roughness of both (Fig. 7b); but these gaps (and the free electrolyte in the gap) are significantly fewer than the case of UiO-66@PP separator. In the closely contacted section (of Zr-MOCN@PP and Li⁰-electrode), the partially desolvated Li⁺-species are more easily reduced to Li-metal, and thus deposited preferentially, which could have a repair effect to surrounding gaps. Therefore, in the Zr-MOCN@PP based cells, the parasitic reaction cannot be completely ruled out, as we can find signs of by-products from XPS and ToF-SIMS; but their continuous occurrence was effectively suppressed (Fig. 2e,f; Fig. 4)”

Q2: The videos are excellent, however, please provide details on what we're seeing and how the experiment was conducted. Explaining the cell structure with a schematic would be helpful.

We appreciate the reviewer's suggestions. Accordingly, we have added the detailed information for the *in-situ* optical microscopy measurement in the Method section of the revised manuscript (lines 540-548)

“In-situ Optical Microscopy Measurements. *Firstly, the pouch cell was constructed by thin lithium metal as anode, graphite as counter electrode and Zr-MOCN or PP as a*

separator (Supplementary Fig. 20). Then, the cell was clamped and cut carefully to obtain a smooth section surface. After that, a large amount of liquid electrolyte (LiPF₆-LE) was filled through the cross-section. Finally, the sample was assembled into the testing system for observation. It should be noted that several short circuit tests were required during the sample preparation to ensure the cell could charge/discharge well. The graphite anode was charged before observation; followed that the cell discharged at the current density of 1.0 mA cm⁻² to observe the lithium electrodeposition behaviour.”

Moreover, we drew the scheme of the sample preparation and observation as Supplementary Fig. 20 in the revised Supplementary file:

Supplementary Figure 20 | Scheme of the sample preparation for *in-situ* optical microscopy observation.

The images of the Li-deposition morphology observed by *in-situ* optical microscopy were also added as Supplementary Fig. 18, and revised the sentence of the manuscript on **Line 156-164**:

“Moreover, to directly observe the lithium deposition behaviour, an *in-situ* optical microscopy measurement was performed with the pouch cell, which was constructed by

thin lithium metal as anode, graphite as counter electrode, and a separator with an excessive amount of $\text{LiPF}_6\text{-LE}$ (Supplementary Fig. 20). Dendrites were immediately produced and crazily grew when using the PP separator (Supplementary Video 1; Supplementary Fig. 21c,d); while the lithium was smoothly deposited on the Li surface in the Zr-MOCN@PP cell (Supplementary Video 1; Supplementary Fig. 21a,b), and no dendrite was observed. This result provides additional confirmation for the argument that Zr-MOCN@PP membrane effectively eliminated Li^0 dendrite formation.”

Supplementary Figure 21 | Observation of Li-deposition morphology by *in-situ* optical microscopy. **a, b**, The Zr-MOCN@PP based cell discharged at a current density of 1.0 mA cm^{-2} after 500 s and 2000 s, respectively. **c, d**, The PP separator based cell discharged at a current density of 1.0 mA cm^{-2} after 500 s and 2000 s, respectively.

Q3: P-Zr-MOC has pore size between 1.41 – 2.77nm. UiO-66 has pore size of 0.7 – 0.9nm, which is smaller than a fully solvated Li^+ as cited by the authors (line 84). Please explain in more detail why modeling efforts with UiO-66 are applicable to Zr-MOCN. How does pore size affect surface interaction and how does fully solvated Li^+ fit into smaller UiO-66 pore?

We appreciate the reviewer’s comments. The Li^+ -solvation species, such as $\text{Li}(\text{EC})_4^+$ has a size of $\sim 1 \text{ nm}$, a DFT calculated value that is often used in the electrolyte structure papers. The pore size of 0.7-0.9 nm for UiO-66 in the manuscript, it’s an experimental value that was analyzed

by N₂-sorption isotherm measurement. These two values are close, and both are correct in their environments but they came from different sources; thus here we compared them using DFT calculated values. As we can see in the following Fig. 1, people used some equivalent diameters to describe the size of molecules or pores, but for actual molecules, they have irregular shapes rather than simple spheres. For Li(EC)₄⁺, it has a methane-type tetrahedron structure. The distance from its vertex to the bottom plane is about 7.7 Å, and also has vertex-to-vertex distances from 8.6 to 10.8 Å depending on different vertex atoms. For the UiO-66, the geometry of its pores is also not circular, the size of which is 10.5~14.3Å depending on the different ways of measurement (Fig. 1b, 1c). Therefore, when compared under the same DFT calculated models, the pore size of UiO-66 is slightly larger than that of Li(EC)₄⁺.

Further, it is worth noting that Li(EC)₄⁺ is not a rigid molecule; in fact, it is not a molecule nor even a coordination complex. Its solvation sheath is flexible and prone to deformation, as revealed in the liquid electrolyte molecular dynamic studies^{1,2}. On the other hand, we also utilized the DFT method to calculate the UiO-66 hosted Li(EC)₄⁺ model (Fig. 5c). The framework of the UiO-66 was rendered grey as we wanted to highlight the guest molecule (Li(EC)₄⁺). In this model, the pores of UiO-66 can accommodate Li(EC)₄⁺, and the enlarged figures are also shown in the following Figure 1c.

As shown in Supplementary Fig. 22, the precursor of the Zr-MOCN, its structure was very similar to that of UiO-66. Both of them have a Zr₆O₄(OH)₄ core and 12 carboxylate ligands. Their almost identical structures also give them similar pore environments, and it is the main reason we use UiO-66 to model the amorphous Zr-MOCN. Because the desolvation of guest solvates in the pores, is related to the interaction between the guest molecules and porous matrix. Moreover, although the Zr-MOC owns a well-defined structure (Fig. 1b) (and we also obtained it single crystal), it transformed to be amorphous network after photo-polymerization (Supplementary Fig. 24).

For such an inorganic-organic composite that contains many transition metals, constructing a model and studying its interaction with guest molecules using current computational chemistry tools will be very challenging to get some reliable conclusions even if it is possible. Thus, we use the highly symmetrical structure of UiO-66 to simplify the model. Moreover, we also experimentally studied the electrolyte filled UiO-66, such as their desolvation and Li⁺-electrodeposition behaviour, and found it has a positive effect on suppressing dendrite and parasitic reactions.

As for the effect of pore size on Li⁺-solvation structures, we could not give a clear conclusion here. However, in our previously studied system³ we have investigated the Li⁺-solvation structures

in much larger pores (with a size of ~ 3.3 nm); and found the surface adsorption assisted desolvation of $\text{Li}(\text{EC})_4^+$.

1. Allen, J. L., Borodin, O., Seo, D. M. & Henderson, W. A. Combined quantum chemical/Raman spectroscopic analyses of Li^+ cation solvation: Cyclic carbonate solvents—Ethylene carbonate and propylene carbonate. *J. Power Sources* 267, 821-830 (2014).
2. Suo, L. et al. “Water-in-salt” electrolyte enables high-voltage aqueous lithium-ion chemistries. *Science* 350, 938-943 (2015).
3. Sheng, L. et al. Accelerated Lithium-ion Conduction in Covalent Organic Frameworks. *Chem. Commun.* 56, 10465-10468 (2020).

Figure 1. **a**, The size of the $\text{Li}(\text{EC})_4^+$. **b, c**, The size of the UiO-66. **d, e, f**, The view of $\text{Li}(\text{EC})_4^+$ in the pore of the UiO-66 from different directions.

Therefore, some descriptions about we chose UiO-66 as model were also added in the revised manuscript on Line 235-243:

“As for the amorphous feature of the polymerized Zr-MOC (Supplementary Fig. 24), studying the host-guest interaction for such transition-metal contained large scale system using current computational chemistry tools is quite challenging. Therefore, to simplify the porous structure of Zr-MOCN and save the computational cost, a highly symmetrical metal-organic framework (MOF) analogue, UiO-66 (Supplementary Fig. 25) was chosen as the host materials for modelling. Since the precursor of Zr-MOCN and UiO-66 consist of a $\text{Zr}_6\text{O}_4(\text{OH})_4$ core and 12 carboxylate ligands, the structural similarity of Zr-MOCN and UiO-66 give them similar pore environment, which is one important factor for the host-guest interactions.”

Q4: How does VC additive factor into all of this? Won't it preferentially decompose on Li regardless?

We appreciate the reviewer's comments. The liquid electrolyte composed of 1.0 M LiPF₆ in EC/DMC/EMC (1/1/1, w/w/w) and 1 wt% of VC is commonly used in the industry, and it is also the most commonly used electrolyte in our lab. For comparison, we have also assembled a Li-symmetric cell that uses 1.0 M LiPF₆ in EC/DMC/EMC (1/1/1, w/w/w) as a liquid electrolyte for measurement. As shown in the following Fig. 2a, although the liquid electrolyte did not add with VC additive, the Zr-MOCN@PP membrane exhibited much lower voltage polarization than the PP separator. Generally, vinylene carbonate (VC) is a promising SEI forming additive (via its decomposition) for graphite anode, which could inhibit the further decomposition of the liquid electrolyte and prevent the co-intercalation of the Li⁺ and solvent molecules in the lithium-ion batteries. The lithium metal cell (here was Li-symmetric cell) that using PP separator showed a little better cycling performance after adding the VC additive (Fig. 2b, 2c); however, in the Zr-MOCN based cell, the VC did not show obvious difference on the cycling performance (Fig. 2c).

Figure 2. Electrochemical performance in Li-symmetric cells. Initial voltage profiles of: **a**, the PP separator and Zr-MOCN@PP-based cells at a fixed current density of 1 mA cm⁻² using different liquid electrolytes without VC (1 M LiPF₆ in EC/DMC/EMC, 1/1/1, w/w/w). **b**, PP separator based cell with VC (1 M LiPF₆ in

EC/DMC/EMC, 1/1/1, w/w/w and 1 wt% VC) and without VC. c, PP separator and Zr-MOCN@PP based cell with and without VC.

Q5: (line 460) Please provide areal capacity (mAh/cm²) of the cathode so that mA/cm² can be calculated using reported c-rates. This information would be helpful to understand why Li deposits are an order of magnitude larger in the full cell than in the symmetric cells (Figure 2 f versus m). Information such as electrolyte loading, n/p ratio, and cell stack pressure is also critical information to include

We appreciate the reviewer's suggestions, which helped us improve the quality of our work. Accordingly, we have added the detailed information of the electrode preparation and cell assembling in the Electrochemical Measurements of the Methods section on **Line 532-537**:

“The active materials' weight density for an NMC622 electrode sheet is 4.5 mg cm⁻², and the areal capacity loading is 0.855 mAh cm⁻². 1.0 M LiPF₆ in EC/DMC/EMC (1/1/1, w/w/w) and 1 wt% of VC was used as a liquid electrolyte. For assembling a lithium metal full cell, the NMC622 with the diameter of 12 mm was used as a cathode, the lithium metal foil with the thickness of around 400 μm and the diameter of 15.4 mm was used as an anode (used as received without any treatment). The pressure was 50 kg cm⁻² for cell stack.”

Q6: (Figure 2) It's surprising that filling the Celgard pores with a MOCN reduces the resistance of the separator. EIS testing of the symmetric cells would be useful to differentiate between R_{ct} of the Li electrode and the Ohmic resistance of the separator. The authors

We appreciate the reviewer's suggestions, which helped us improve the quality of our work. Accordingly, we have carried out the EIS measurements of the Li-symmetric cells with Zr-MOCN@PP and PP separator. The discussion was added in the revised manuscript on **Lines 132-136**:

“From electrochemical impedance spectroscopy (EIS) (Supplementary Fig. 12), before cycling (after 1 h of cell assembly), the Zr-MOCN@PP based cell demonstrated a slightly higher Ohmic resistance (R_Ω), but a significant lower SEI resistance (R_{SEI}) than the pristine PP separator based cell. And for the cycled cells, the R_{SEI} of Zr-MOCN@PP kept at a low value, while the R_{SEI} of PP largely increased from 10th cycle to 200th cycle.”

The detailed information for the impedance measurement was also added in the revised Supplementary Information:

“Impedance analyses of the Li-symmetric cells with PP separator and Zr-MOCN@PP were performed on a CHI660E Electrochemical Workstation (Shanghai Chenhua) with electrochemical impedance spectroscopy (EIS). The perturbation amplitude was 10 mV and the frequency range from 0.1 Hz to 10 kHz at room temperature.”

Supplementary Fig. 12 was added in the revised Supplementary file:

Supplementary Figure 12 | Impedance spectroscopy of Li-symmetric cells with the PP separator and Zr-MOCN@PP before and after stripping/plating at a current density of 1 mA cm⁻², respectively. **a**, Before stripping/plating. **b**, **c**, after 10th, 200th stripping/plating of the PP separator and Zr-MOCN@PP based cells, respectively.

Reviewer 2:

The authors have reported that the direct suppressing electrolyte and lithium metal reactivity by producing nano-structured photoresist membrane of porous Zr-MOC. They found out the polymerized Zr-MOC membrane was enough to stable and dense for protecting Li metal surface from Li dendrite formation with DFT calculation of the suppressing mechanism. It is interesting to study the inhibition of dendrite formation in Lithium-metal batteries by synthesizing nano-porous separators using photoresist. However, the structural analysis of Zr-MOC and Zr-MOCN was insufficient, and the flow of the overall paper was cluttered. It is interesting to calculate the mechanism that inhibits dendrite formation, but it is questionable whether the UiO-66 as a model is appropriate. It is recommended to reject this paper in current form and the following items are detailed comments.

We appreciate the reviewer's suggestions and comments, which helped us to improve the quality of our work.

Q1: The labeling of the research product was insufficient throughout the thesis. It may reduce the reader's understanding of this article. For example, lines 82-86 on page 4 describe the BET surface area. The first indication was that the polymerized Zr-MOC was p-Zr-MOC, but the relevant supplementary data was Zr-MOCN. In addition, the notation of Zr-MOCN began in the next paragraph. Therefore, it is recommended to clearly denote the product.

We appreciate the reviewer's suggestions, which helped us to improve the quality of our work. Accordingly, we have checked and re-denoted the products in the manuscript to give a clear picture of their characteristics. Now, abbreviations for all products involved are as follows:

Zr-MOC: Zr-contained metal-organic cluster (**unchanged**).

Zr-MOCN: Zr-contained metal-organic cluster network (photopolymerized Zr-MOC without supporter, **changed from p-Zr-MOC**).

Zr-MOC@PP: pristine Zr-MOC at PP substrate (**unchanged**).

Zr-MOCN@PP: *in-situ* photopolymerized Zr-MOC at PP substrate (**changed from Zr-MOCN**).

Q2: This is the same as point 1. Please clarify the structural explanations for $Zr_6O_4(OH)_4(MMA)_{12}$ cluster or p-ZrMOC on page 6 in line 138.

We appreciate the reviewer's suggestions, which helped us to improve the quality of our work. The " $Zr_6O_4(OH)_4(MMA)_{12}$ cluster" in the original manuscript referred to the pristine Zr-contained

metal-organic cluster, while “p-ZrMOC” referred to polymerized Zr-MOC. Apologize for the confusion brought by our previous abbreviations. As suggested by the reviewer in Q1, we re-denoted these two products, and now the sentence was revised as follows (Line 144-145):

“To differentiate whether the reduced Li^0 reactivity arises from the Zr-MOC (pristine $Zr_6O_4(OH)_4(MAA)_{12}$ cluster) or Zr-MOCN.....”

Q3: The synthesized Zr-MOC was described as a crystal structure, but there was no evidence to support it, such as an x-ray diffraction pattern. The CCDC number mentioned was insufficient and supplemental data are required.

We appreciate the reviewer’s suggestions and comments. Accordingly, we have added detailed information for the obtained Zr-MOC single crystal as Supplementary Table 1-5. Meanwhile, as for the X-ray diffraction pattern mentioned by reviewer, we have also characterized the Zr-MOC crystals using powder X-ray diffraction, and appended the data in Supplementary Fig. 2:

Supplementary Table 1 | Crystal data and structure refinement for Zr-MOC (CCDC No: 2022033)

Identification code	Zr-MOC (CCDC No: 2022033)	
Empirical formula	C ₄₈ H ₈₄ O ₃₂ Zr ₆	
Formula weight	1720.43	
Temperature	173.00(10) K	
Wavelength	0.71073 Å	
Crystal system	Hexagonal	
Space group	P6 ₃ mc	
Unit cell dimensions	a = 17.3180(8) Å	a = 90°.
	b = 17.3180(8) Å	b = 90°.
	c = 18.1144(6) Å	g = 120°.
Volume	4704.9(5) Å ³	
Z	2.00004	
Density (calculated)	1.214 Mg/m ³	
Absorption coefficient	0.700 mm ⁻¹	
F(000)	1736	
Crystal size	0.45 x 0.35 x 0.2 mm ³	
Theta range for data collection	3.255 to 29.633°.	
Index ranges	-23<=h<=23, -19<=k<=22, -25<=l<=24	

Reflections collected	45260
Independent reflections	4590 [R(int) = 0.0706]
Completeness to theta = 25.242°	99.6 %
Absorption correction	Semi-empirical from equivalents
Max. and min. transmission	1.00000 and 0.71762
Refinement method	Full-matrix least-squares on F ²
Data / restraints / parameters	4590 / 124 / 213
Goodness-of-fit on F ²	1.026
Final R indices [I>2sigma(I)]	R1 = 0.0599, wR2 = 0.1428
R indices (all data)	R1 = 0.0985, wR2 = 0.1620
Absolute structure parameter	-0.09(3)
Extinction coefficient	n/a
Largest diff. peak and hole	0.517 and -0.447 e.Å ⁻³

Supplementary Table 2 | Atomic coordinates ($\times 10^4$) and equivalent isotropic displacement parameters ($\text{\AA}^2 \times 10^3$) for Zr-MOC (CCDC No: 2022033). U(eq) is defined as one third of the trace of the orthogonalized U^{ij} tensor.

	x	y	z	U(eq)
Zr(01)	7346(1)	2654(1)	4918(1)	63(1)
Zr(02)	5993(1)	1985(1)	3360(1)	68(1)
O(003)	6667	3333	3063(6)	54(3)
O(004)	6098(3)	2197(5)	4490(4)	61(2)
O(005)	8185(7)	3336(11)	5844(5)	167(5)
O(006)	5944(14)	773(18)	3681(11)	82(5)
O(007)	7420(3)	2580(3)	3615(5)	81(3)
O(008)	6667	3333	5482(5)	59(3)
O(009)	6858(16)	1185(18)	4671(14)	101(7)
O(00A)	6476(7)	1699(6)	2279(4)	110(3)
C(00C)	6382(18)	670(16)	4073(14)	89(5)
C(00E)	5715(12)	1430(20)	1113(14)	177(7)
C(00H)	5807(9)	1614(19)	1939(10)	140(5)
C(00I)	6833(18)	-397(18)	4729(18)	125(9)
C(00K)	6333(18)	1440(20)	700(10)	237(11)
C(00N)	8440(20)	4222(11)	6085(15)	203(9)
C(5)	6277(19)	-277(15)	4172(16)	106(6)
C(6)	9160(20)	4580(10)	6563(15)	219(9)

C(12)	9510(40)	4150(30)	7050(20)	232(12)
O(14)	7208(14)	1392(16)	4596(12)	80(5)
O(17)	6336(13)	907(16)	3616(11)	80(5)
C(22)	6846(16)	846(15)	4168(14)	80(5)
C(37)	6840(20)	34(15)	4234(17)	109(6)
C(45)	6130(20)	-756(17)	3823(19)	128(10)
C(20)	7460(20)	-50(20)	4724(19)	137(9)
C(46)	5540(20)	-1040(20)	3787(18)	138(11)
C(31)	9370(40)	3850(30)	6770(20)	230(12)

Supplementary Table 3 | Bond lengths [\AA] and angles [$^\circ$] for Zr-MOC (CCDC No: 2022033).

Zr(01)-Zr(02)	3.4762(11)
Zr(01)-Zr(02)#1	3.4762(11)
Zr(01)-O(004)#1	2.047(4)
Zr(01)-O(004)	2.047(4)
Zr(01)-O(005)	2.145(9)
Zr(01)-O(005)#2	2.145(9)
Zr(01)-O(007)	2.370(10)
Zr(01)-O(008)	2.281(4)
Zr(01)-O(009)#2	2.29(3)
Zr(01)-O(009)	2.29(3)
Zr(01)-O(14)	2.16(3)
Zr(01)-O(14)#2	2.16(3)
Zr(02)-Zr(01)#3	3.4762(11)
Zr(02)-O(003)	2.092(3)
Zr(02)-O(004)	2.070(7)
Zr(02)-O(006)#4	2.14(3)
Zr(02)-O(006)	2.14(3)
Zr(02)-O(007)	2.199(6)
Zr(02)-O(007)#3	2.199(5)
Zr(02)-O(00A)#4	2.279(7)
Zr(02)-O(00A)	2.279(7)
Zr(02)-C(00H)	2.633(19)
Zr(02)-O(17)#4	2.27(2)
Zr(02)-O(17)	2.27(2)
O(003)-Zr(02)#3	2.092(3)
O(003)-Zr(02)#1	2.092(3)

O(004)-Zr(01)#3	2.047(4)
O(005)-C(00N)	1.435(16)
O(006)-C(00C)	1.12(2)
O(007)-Zr(02)#1	2.199(5)
O(008)-Zr(01)#1	2.281(4)
O(008)-Zr(01)#3	2.281(4)
O(009)-C(00C)	1.38(2)
O(00A)-C(00H)	1.254(12)
C(00C)-C(5)	1.57(2)
C(00E)-C(00H)	1.52(3)
C(00E)-C(00K)#4	1.30(2)
C(00E)-C(00K)	1.30(2)
C(00H)-O(00A)#4	1.254(12)
C(00I)-H(00A)	0.9600
C(00I)-H(00B)	0.9600
C(00I)-H(00C)	0.9600
C(00I)-C(5)	1.48(2)
C(00K)-H(00D)	0.9600
C(00K)-H(00E)	0.9600
C(00K)-H(00F)	0.9600
C(00N)-O(005)#5	1.435(16)
C(00N)-C(6)	1.38(4)
C(5)-H(5)	0.9800
C(5)-C(46)	1.47(2)
C(6)-C(12)#5	1.47(2)
C(6)-C(12)	1.47(2)
C(6)-C(31)#5	1.53(3)
C(6)-C(31)	1.53(2)
C(12)-C(31)	0.68(6)
O(14)-C(22)	1.14(2)
O(17)-C(22)	1.37(2)
C(22)-C(37)	1.40(2)
C(37)-H(37)	0.9800
C(37)-C(45)	1.50(2)
C(37)-C(20)	1.47(2)
C(45)-H(45A)	0.9600
C(45)-H(45B)	0.9600
C(45)-H(45C)	0.9600
C(20)-H(20A)	0.9600
C(20)-H(20B)	0.9600

C(20)-H(20C)	0.9600
C(46)-H(46A)	0.9600
C(46)-H(46B)	0.9600
C(46)-H(46C)	0.9600
Zr(02)#1-Zr(01)-Zr(02)	60.48(4)
O(004)#1-Zr(01)-Zr(02)#1	32.61(19)
O(004)-Zr(01)-Zr(02)	32.61(19)
O(004)-Zr(01)-Zr(02)#1	83.3(2)
O(004)#1-Zr(01)-Zr(02)	83.3(2)
O(004)#1-Zr(01)-O(004)	92.3(5)
O(004)-Zr(01)-O(005)	143.4(4)
O(004)#1-Zr(01)-O(005)	85.2(5)
O(004)#1-Zr(01)-O(005)#2	143.4(4)
O(004)-Zr(01)-O(005)#2	85.2(5)
O(004)-Zr(01)-O(007)	71.1(3)
O(004)#1-Zr(01)-O(007)	71.1(3)
O(004)-Zr(01)-O(008)	69.6(2)
O(004)#1-Zr(01)-O(008)	69.6(2)
O(004)#1-Zr(01)-O(009)#2	276.3(8)
O(004)-Zr(01)-O(009)	76.3(8)
O(004)#1-Zr(01)-O(009)	146.4(7)
O(004)-Zr(01)-O(009)#2	146.4(7)
O(004)-Zr(01)-O(14)#2	139.1(5)
O(004)-Zr(01)-O(14)	86.8(7)
O(004)#1-Zr(01)-O(14)#2	86.8(7)
O(004)#1-Zr(01)-O(14)	139.1(5)
O(005)#2-Zr(01)-Zr(02)	111.0(4)
O(005)-Zr(01)-Zr(02)	167.4(4)
O(005)-Zr(01)-Zr(02)#1	111.0(4)
O(005)#2-Zr(01)-Zr(02)#1	167.4(5)
O(005)-Zr(01)-O(005)#2	75.7(7)
O(005)#2-Zr(01)-O(007)	140.2(3)
O(005)-Zr(01)-O(007)	140.2(3)
O(005)-Zr(01)-O(008)	75.4(4)
O(005)#2-Zr(01)-O(008)	75.4(4)
O(005)#2-Zr(01)-O(009)#2	2122.5(9)
O(005)-Zr(01)-O(009)	122.5(9)
O(005)#2-Zr(01)-O(009)	68.2(6)
O(005)-Zr(01)-O(009)#2	68.2(6)

O(005)-Zr(01)-O(14)#2 77.3(6)
O(005)-Zr(01)-O(14) 118.1(8)
O(005)#2-Zr(01)-O(14)#2 118.1(8)
O(005)#2-Zr(01)-O(14) 77.3(6)
O(007)-Zr(01)-Zr(02)#1 38.68(13)
O(007)-Zr(01)-Zr(02) 38.68(13)
O(008)-Zr(01)-Zr(02) 95.70(19)
O(008)-Zr(01)-Zr(02)#1 95.70(19)
O(008)-Zr(01)-O(007) 122.0(3)
O(008)-Zr(01)-O(009)#2 131.5(6)
O(008)-Zr(01)-O(009) 131.5(6)
O(009)#2-Zr(01)-Zr(02)#1 70.0(8)
O(009)#2-Zr(01)-Zr(02) 113.8(6)
O(009)-Zr(01)-Zr(02)#1 113.8(6)
O(009)-Zr(01)-Zr(02) 70.0(8)
O(009)-Zr(01)-O(007) 75.2(6)
O(009)#2-Zr(01)-O(007) 75.2(6)
O(009)#2-Zr(01)-O(009) 95.6(13)
O(14)-Zr(01)-Zr(02)#1 107.2(5)
O(14)#2-Zr(01)-Zr(02)#1 74.2(6)
O(14)-Zr(01)-Zr(02) 74.2(6)
O(14)#2-Zr(01)-Zr(02) 107.2(5)
O(14)#2-Zr(01)-O(007) 70.0(6)
O(14)-Zr(01)-O(007) 70.0(6)
O(14)-Zr(01)-O(008) 145.1(5)
O(14)#2-Zr(01)-O(008) 145.1(5)
O(14)#2-Zr(01)-O(14) 68.5(11)
O(003)-Zr(02)-Zr(01)#3 86.0(3)
O(003)-Zr(02)-O(006) 153.0(6)
O(003)-Zr(02)-O(006)#4 153.0(6)
O(003)-Zr(02)-O(007) 70.3(2)
O(003)-Zr(02)-O(007)#3 70.3(2)
O(003)-Zr(02)-O(00A) 87.8(4)
O(003)-Zr(02)-O(00A)#4 87.8(4)
O(003)-Zr(02)-C(00H) 87.3(7)
O(003)-Zr(02)-O(17)#4 137.7(5)
O(003)-Zr(02)-O(17) 137.7(5)
O(004)-Zr(02)-Zr(01)#3 32.20(7)
O(004)-Zr(02)-O(003) 96.1(4)
O(004)-Zr(02)-O(006)#4 82.0(6)

O(004)-Zr(02)-O(006) 82.0(6)
O(004)-Zr(02)-O(007) 74.4(3)
O(004)-Zr(02)-O(007)#3 74.4(3)
O(004)-Zr(02)-O(00A)#4 151.4(3)
O(004)-Zr(02)-O(00A) 151.4(3)
O(004)-Zr(02)-C(00H) 176.6(7)
O(004)-Zr(02)-O(17) 84.7(6)
O(004)-Zr(02)-O(17)#4 84.7(6)
O(006)-Zr(02)-Zr(01)#3 104.7(5)
O(006)#4-Zr(02)-Zr(01)#3 78.2(6)
O(006)#4-Zr(02)-O(006) 53.6(11)
O(006)#4-Zr(02)-O(007)#3 383.5(6)
O(006)-Zr(02)-O(007)#3 133.6(5)
O(006)-Zr(02)-O(007) 83.5(6)
O(006)#4-Zr(02)-O(007) 133.6(5)
O(006)-Zr(02)-O(00A) 81.9(6)
O(006)#4-Zr(02)-O(00A)#4 481.9(6)
O(006)-Zr(02)-O(00A)#4 106.7(7)
O(006)#4-Zr(02)-O(00A) 106.7(7)
O(006)#4-Zr(02)-C(00H) 94.9(8)
O(006)-Zr(02)-C(00H) 94.9(8)
O(007)-Zr(02)-Zr(01)#3 99.5(2)
O(007)#3-Zr(02)-Zr(01)#3 42.3(2)
O(007)#3-Zr(02)-O(007) 125.7(4)
O(007)#3-Zr(02)-O(00A) 132.9(4)
O(007)-Zr(02)-O(00A)#4 132.9(4)
O(007)-Zr(02)-O(00A) 80.4(3)
O(007)#3-Zr(02)-O(00A)#4 480.4(3)
O(007)-Zr(02)-C(00H) 106.9(4)
O(007)#3-Zr(02)-C(00H) 106.9(4)
O(007)#3-Zr(02)-O(17)#4 69.4(6)
O(007)-Zr(02)-O(17)#4 147.6(5)
O(007)#3-Zr(02)-O(17) 147.6(5)
O(007)-Zr(02)-O(17) 69.4(6)
O(00A)-Zr(02)-Zr(01)#3 173.4(2)
O(00A)#4-Zr(02)-Zr(01)#3 120.7(3)
O(00A)#4-Zr(02)-O(00A) 56.8(5)
O(00A)-Zr(02)-C(00H) 28.4(3)
O(00A)#4-Zr(02)-C(00H) 28.4(3)
C(00H)-Zr(02)-Zr(01)#3 148.68(13)

O(17)#4-Zr(02)-Zr(01)#3 72.4(6)
O(17)-Zr(02)-Zr(01)#3 112.4(5)
O(17)-Zr(02)-O(00A)#4 111.2(6)
O(17)#4-Zr(02)-O(00A) 111.2(6)
O(17)#4-Zr(02)-O(00A)#4 73.8(5)
O(17)-Zr(02)-O(00A) 73.8(5)
O(17)-Zr(02)-C(00H) 92.8(7)
O(17)#4-Zr(02)-C(00H) 92.8(7)
O(17)#4-Zr(02)-O(17) 84.5(10)
Zr(02)-O(003)-Zr(02)#3 113.6(2)
Zr(02)#1-O(003)-Zr(02)#3 113.6(2)
Zr(02)-O(003)-Zr(02)#1 113.6(2)
Zr(01)-O(004)-Zr(01)#3 119.2(3)
Zr(01)-O(004)-Zr(02) 115.2(2)
Zr(01)#3-O(004)-Zr(02) 115.2(2)
C(00N)-O(005)-Zr(01) 126.2(15)
C(00C)-O(006)-Zr(02) 129(2)
Zr(02)#1-O(007)-Zr(01) 99.0(3)
Zr(02)-O(007)-Zr(01) 99.0(3)
Zr(02)#1-O(007)-Zr(02) 105.5(4)
Zr(01)-O(008)-Zr(01)#3 101.5(3)
Zr(01)-O(008)-Zr(01)#1 101.5(3)
Zr(01)#3-O(008)-Zr(01)#1 101.5(3)
C(00C)-O(009)-Zr(01) 130(2)
C(00H)-O(00A)-Zr(02) 91.7(9)
O(006)-C(00C)-O(009) 129(2)
O(006)-C(00C)-C(5) 121(2)
O(009)-C(00C)-C(5) 108(2)
C(00K)#4-C(00E)-C(00H) 124.3(15)
C(00K)-C(00E)-C(00H) 124.3(15)
C(00K)#4-C(00E)-C(00K) 110(3)
O(00A)#4-C(00H)-Zr(02) 59.9(8)
O(00A)-C(00H)-Zr(02) 59.9(8)
O(00A)#4-C(00H)-O(00A) 119.8(17)
O(00A)-C(00H)-C(00E) 120.1(8)
O(00A)#4-C(00H)-C(00E) 120.1(8)
C(00E)-C(00H)-Zr(02) 178(2)
H(00A)-C(00I)-H(00B) 109.5
H(00A)-C(00I)-H(00C) 109.5
H(00B)-C(00I)-H(00C) 109.5

C(5)-C(00I)-H(00A)	109.5
C(5)-C(00I)-H(00B)	109.5
C(5)-C(00I)-H(00C)	109.5
C(00E)-C(00K)-H(00D)	109.5
C(00E)-C(00K)-H(00E)	109.5
C(00E)-C(00K)-H(00F)	109.5
H(00D)-C(00K)-H(00E)	109.5
H(00D)-C(00K)-H(00F)	109.5
H(00E)-C(00K)-H(00F)	109.5
O(005)-C(00N)-O(005)#5	132(2)
C(6)-C(00N)-O(005)	113.7(12)
C(6)-C(00N)-O(005)#5	113.7(12)
C(00C)-C(5)-H(5)	92.6
C(00I)-C(5)-C(00C)	119(2)
C(00I)-C(5)-H(5)	92.6
C(46)-C(5)-C(00C)	118(2)
C(46)-C(5)-C(00I)	122(2)
C(46)-C(5)-H(5)	92.6
C(00N)-C(6)-C(12)#5	131(2)
C(00N)-C(6)-C(12)	131(2)
C(00N)-C(6)-C(31)#5	108.6(19)
C(00N)-C(6)-C(31)	108.6(19)
C(12)#5-C(6)-C(12)	91(3)
C(12)#5-C(6)-C(31)#5	26(2)
C(12)-C(6)-C(31)	26(2)
C(12)-C(6)-C(31)#5	117(2)
C(12)#5-C(6)-C(31)	117(2)
C(31)-C(6)-C(31)#5	142(3)
C(31)-C(12)-C(6)	82(3)
C(22)-O(14)-Zr(01)	140(2)
C(22)-O(17)-Zr(02)	130.9(16)
O(14)-C(22)-O(17)	123(2)
O(14)-C(22)-C(37)	120(2)
O(17)-C(22)-C(37)	117(2)
C(22)-C(37)-H(37)	91.0
C(22)-C(37)-C(45)	118(2)
C(22)-C(37)-C(20)	121(2)
C(45)-C(37)-H(37)	91.0
C(20)-C(37)-H(37)	91.0
C(20)-C(37)-C(45)	121.2(19)

C(37)-C(45)-H(45A)	109.5
C(37)-C(45)-H(45B)	109.5
C(37)-C(45)-H(45C)	109.5
H(45A)-C(45)-H(45B)	109.5
H(45A)-C(45)-H(45C)	109.5
H(45B)-C(45)-H(45C)	109.5
C(37)-C(20)-H(20A)	109.5
C(37)-C(20)-H(20B)	109.5
C(37)-C(20)-H(20C)	109.5
H(20A)-C(20)-H(20B)	109.5
H(20A)-C(20)-H(20C)	109.5
H(20B)-C(20)-H(20C)	109.5
C(5)-C(46)-H(46A)	109.5
C(5)-C(46)-H(46B)	109.5
C(5)-C(46)-H(46C)	109.5
H(46A)-C(46)-H(46B)	109.5
H(46A)-C(46)-H(46C)	109.5
H(46B)-C(46)-H(46C)	109.5
C(12)-C(31)-C(6)	72(3)

Symmetry transformations used to generate equivalent atoms:

#1 -y+1,x-y,z #2 -y+1,-x+1,z #3 -x+y+1,-x+1,z
 #4 -x+y+1,y,z #5 x,x-y,z

Supplementary Table 4 | Anisotropic displacement parameters ($\text{\AA}^2 \times 10^3$) for Zr-MOC (CCDC No: 2022033). The anisotropic displacement factor exponent takes the form: $-2p^2 [h^2 a^*^2 U^{11} + \dots + 2 h k a^* b^* U^{12}]$

	U11	U22	U33	U23	U13	U12
Zr(01)	87(1)	87(1)	36(1)	2(1)	-2(1)	58(1)
Zr(02)	96(1)	65(1)	33(1)	-10(1)	-5(1)	33(1)
O(003)	59(4)	59(4)	44(6)	0	0	29(2)
O(004)	76(4)	65(4)	39(4)	-15(3)	-7(2)	33(2)
O(005)	117(6)	317(16)	65(5)	-14(7)	-32(5)	105(8)
O(006)	113(13)	90(11)	55(8)	-1(7)	15(9)	59(12)
O(007)	107(6)	107(6)	56(5)	-2(2)	2(2)	73(6)
O(008)	77(5)	77(5)	23(5)	0	0	39(3)
O(009)	138(15)	94(12)	87(11)	23(9)	49(10)	71(11)

O(00A)	167(7)	123(6)	51(4)	-23(4)	7(4)	80(6)
C(00C)	125(12)	82(10)	81(9)	-2(8)	32(10)	68(10)
C(00E)	224(15)	209(14)	93(10)	-26(12)	-13(6)	105(7)
C(00H)	188(11)	168(10)	56(8)	-16(9)	-8(4)	84(5)
C(00I)	125(18)	101(16)	170(20)	8(16)	-7(18)	73(15)
C(00K)	280(20)	280(20)	108(13)	-44(16)	2(14)	109(18)
C(00N)	138(11)	325(19)	85(10)	-13(4)	-25(9)	69(6)
C(5)	136(14)	84(12)	106(12)	-5(10)	21(12)	59(12)
C(6)	145(12)	340(20)	105(12)	-10(5)	-20(10)	72(6)
C(12)	153(15)	350(20)	115(16)	1(12)	-21(14)	69(12)
O(14)	118(13)	90(11)	62(8)	27(7)	39(9)	74(10)
O(17)	109(12)	69(9)	62(8)	8(6)	41(9)	43(10)
C(22)	117(12)	71(9)	74(8)	15(7)	17(9)	65(9)
C(37)	136(14)	88(11)	104(11)	13(10)	8(12)	56(11)
C(45)	140(20)	69(14)	150(20)	0(14)	0(20)	39(16)
C(20)	160(20)	95(15)	180(20)	-4(17)	0(20)	81(15)
C(46)	140(20)	130(20)	130(20)	-4(17)	17(19)	48(18)
C(31)	150(15)	350(20)	112(16)	-4(12)	-26(14)	67(12)

Supplementary Table 5 | Hydrogen coordinates ($\times 10^4$) and isotropic displacement parameters ($\text{\AA}^2 \times 10^3$) for Zr-MOC (CCDC No: 2022033).

	x	y	z	U(eq)
H(00A)	7186	-610	4492	188
H(00B)	6454	-823	5092	188
H(00C)	7218	163	4965	188
H(00D)	6865	1630	987	356
H(00E)	6458	1838	294	356
H(00F)	6133	847	515	356
H(5)	6701	-177	3774	128
H(37)	7248	188	3817	131
H(45A)	5739	-593	3579	192
H(45B)	5791	-1232	4164	192
H(45C)	6404	-948	3463	192
H(20A)	8023	500	4725	205
H(20B)	7554	-523	4553	205
H(20C)	7227	-180	5216	205

H(46A)	5108	-880	3634	207
H(46B)	5268	-1541	4113	207
H(46C)	5771	-1187	3361	207

The measurement method added in the Materials Characterization section on **Lines 519-521**:

“Powder X-ray diffraction (PXRD) data was performed on a D8 Advance Brooker Ultima III diffractometer from $2\theta = 4.0^\circ$ up to 60° with a step size of 0.02° and 0.5 s per step.”

Supplementary Figure 2 | PXRD profiles of Zr-MOC (red: experimentally observed; green: Pawley refined; black: their difference; blue: theoretical data) with $R_{wp} = 6.71\%$ and $R_p = 4.60\%$.

Q4: In Figures 2a and b, there were no explicit polarization values are mentioned as in Figure 6c (eg. 60-100 mV, 240-400 mV). It would be helpful to compare each data by complementing exact values.

We appreciate the reviewer’s suggestions and comments. Accordingly, we have added the

exact polarization values for Figures 2a and b in the sentences as follows (Line 112-120):

“Zr-MOCN@PP membrane exhibited apparently reduced voltage polarization (around 10 mV) than PP (80 mV) in the initial several cycling (Fig. 2a) at a current density of 1 mA cm⁻² (the areal capacity was 1 mAh cm⁻²). Such cell could be cycled for more than 2000 h with the voltage profiles remaining essentially unchanged.....”

In sharp contrast, the cell containing LiPF₆-LE in PP displays a cell polarization increased largely (around 400 mV) that significantly deteriorates, and the cell terminates with an obvious short circuit at the 270th hour (Fig. 2a).”

Q5: In Figure 4b and d, it is recommended to add the scale of the ToF-SIMS data.

We appreciate the reviewer’s suggestions, which helped us to improve the quality of our work. We have added the scale of the ToF-SIMS data in the revised Supplementary file as Supplementary Fig. 23a, b. For better understanding, the 3D mapping images of several typical second ion fragments based on the ToF-SIMS depth scan were also added (Supplementary Fig. 23d, e).

Supplementary Figure 23 | Characterization of chemical compositions on the cycled lithium metal electrodes by ToF-SIMS. **a, b, c** ToF-SIMS depth profiling of several typical second ion fragments on the cycled

lithium metal electrode surface of the pristine PP and Zr-MOCN protected areas, respectively. **d, e** ToF-SIMS 3D mapping of second ion fragments with 1600 s of sputtering in the lithium metal electrodes cycled in the PP and Zr-MOCN protected areas, respectively.

Q6: The mechanism for parasitic reaction suppression was calculated with DFT method with UiO-66. However, in Supplementary Figure 15, UiO-66 and Zr-MOCN had different structure. The UiO-66 is a particle with continuously arranged pores, but the Zr-MOCN described in this text is a continuously connected membrane by polymerization of Zr-MOC. I recommend to explain of the point at which Zr-MOCN can be structurally simplified to UiO-66.

We appreciate the reviewer's suggestions, which helped us to improve the quality of our work. As shown in Supplementary Fig. 25, the precursor of the Zr-MOCN, its structure was very similar to that of UiO-66. Both of them have a $Zr_6O_4(OH)_4$ core and 12 carboxylate ligands. Their almost identical structures also give them similar pore environments, and it is the main reason we use UiO-66 to model the amorphous Zr-MOCN. Because the desolvation of guest solvates in the pores, is related to the interaction between the guest molecules and porous matrix. Moreover, although the Zr-MOC owns a well-defined structure (Fig. 1c) (and we also obtained its single crystal), it transformed to be amorphous network after photo-polymerization (Supplementary Fig. 24).

For such an inorganic-organic composite that contains many transition metals, constructing a model and studying its interaction with guest molecules using current computational chemistry tools will be very challenging to get some reliable conclusions even if it is possible. Thus, we use the highly symmetrical structure of UiO-66 to simplify the model. Moreover, we also experimentally studied the electrolyte filled UiO-66, such as their desolvation and Li^+ -electrodeposition behaviour, and found it has a positive effect on suppressing dendrite and parasitic reactions.

Therefore, some descriptions about we chose UiO-66 as model were also added in the revised manuscript on **Line 235-243**:

“As for the amorphous feature of the polymerized Zr-MOC (Supplementary Fig. 24), studying the host-guest interaction for such transition-metal contained large scale system using current computational chemistry tools is quite challenging. Therefore, to simplify the porous structure of Zr-MOCN and save the computational cost, a highly symmetrical metal-organic framework (MOF) analogue, UiO-66 (Supplementary Fig. 25) was chosen as the host materials for modelling. Since the precursor of Zr-MOCN and UiO-66 consist of a $Zr_6O_4(OH)_4$ core and 12 carboxylate ligands, the structural

similarity of Zr-MOCN and UiO-66 give them similar pore environment, which is one important factor for the host-guest interactions.”

Q7: Also, there is no detailed description of what arrangement or structure Zr-MOCN actually has in the text.

We appreciate the reviewer’s comments. The polymerized Zr-MOC is a random crosslinked network, the organic ligands (methacrylic acid) can be reacted with each other under the UV light when adding radical initiator. However, there has 12 methacrylic acid (Fig. 1b) in the Zr-MOC structure, random crosslinking has occurred, which also can be seen from the XRD result (Supplementary Fig. 24).

Accordingly, we added the sentences in the revised manuscript on **Line 235**:

“As for the amorphous feature of the Zr-MOCN (Supplementary Fig. 24), ……”

Supplementary Figure 24 | PXRD profiles of Zr-MOCN.

Q8: The preparation of Zr-MOCN membrane and Zr-MOC@PP separator were very different. One used immersion method and the other used doctor blade method. It might have different thickness of Zr-MOC and the use of different binders can lead to different electrochemical reactions. It is recommended to compare the results using a synthetic method similar to Zr-MOCN membrane.

We appreciate the reviewer’s suggestions and comments. Zr-MOC is a powder-like material; thus in this work, the PP supported Zr-MOC membrane (Zr-MOC@PP) was prepared using a similar method with another powder material UiO-66 (corresponding to UiO-66@PP sample). As for the concerns on membrane thickness and binders, according to the reviewer’s suggestion, we

have prepared the PP supported Zr-MOC membrane using a similar way to Zr-MOCN@PP. The detailed preparation method has been added in the Supplementary Information. However, it should be noted that, Zr-MOC has good solubility in the carbonate solvent, such as DMC. Therefore, the Zr-MOC immersed PP membranes were further treated in two different ways:

- 1) Zr-MOC@PP-2, air-dried in the glovebox, then soaked in LiPF₆-liquid electrolyte for 24 h before cell assembling. (exactly same method with Zr-MOCN, except without photopolymerization procedure);
- 2) Zr-MOC@PP-3, air-dried in the glovebox, then directly add electrolyte and cell assembly.

As shown in Supplementary Fig. 18, the performance of the Zr-MOC@PP-2 based Li-symmetric cell was similar to that of the pristine PP separator due to the good solubility of Zr-MOC in the carbonate solvent (Supplementary Fig. 17a). As for the Zr-MOC@PP-3 separator, the introduction of the nonporous Zr-MOC partially blocked pores of the PP substrate (Supplementary Fig. 17b). Therefore, the voltage polarization of the Zr-MOC@PP-3 based cell was higher than the pristine PP separator (Supplementary Fig. 18). Meanwhile, both Zr-MOC@PP-2 and Zr-MOC@PP-3 did not show obvious effect in suppressing lithium dendrite and parasitic reactions (Supplementary Fig. 19).

Accordingly, discussion about these two control experiments have been added in the revised manuscript on **Line 152-156**:

“To rule out the possible effect of thickness and binders of Zr-MOC@PP, we added two additional control experiments using Zr-MOC immersed PP (detailed preparation methods in Supplementary Information). We found both Zr-MOC@PP-2 and Zr-MOC@PP-3 separators showed no obvious effect in suppressing lithium dendrite and parasitic reactions (Supplementary Fig. 17-19).”

Supplementary Figure 17 | SEM images of **a**, Zr-MOC@PP-2; **b**, Zr-MOC@PP-3. The inset is the photos of the Zr-MOC@PP-2 and Zr-MOC@PP-3 membranes, respectively.

Supplementary Figure 18 | Voltage versus time for symmetric lithium cell where each half-cycle lasts 1 h. Initial voltage profiles of the Zr-MOC@PP-2 separator (purple curve) and Zr-MOC@PP-3 (light blue curve)-based cell at a fixed current density of 1.0 mA cm^{-2} (1.0 mAh cm^{-2}).

Supplementary Figure 19 | SEM images of the lithium metal anode after 120 h cycling at 1.0 mA cm^{-2} , 1.0 mAh cm^{-2} in the different separators based symmetric lithium cells. **a, b**, Zr-MOC@PP-2. **c, d**, Zr-MOC@PP-3.

Moreover, the preparation of the Zr-MOC@PP was also added in the revised Supplementary Information:

“Preparation of Zr-MOC@PP-2 separator. The Zr-MOC was dissolved in PGMEA to give a 50 wt/wt% solution. After that, the PP separator was immersed in the mixture,

and the soaked PP separator was taken out, dried in the glovebox. Before assembling a coin cell, the dried separator was immersed in the LiPF₆-LE for 24 h, according to the Zr-MOCN@PP.

Preparation of Zr-MOC@PP-3 separator. *The Zr-MOC was dissolved in PGMEA to give a 50 wt/wt% solution. After that, the PP separator was immersed in the mixture, and the soaked PP separator was taken out, dried in the glovebox. The prepared separators were directly added LiPF₆-LE electrolyte and assembled to a coin cell.”*

Reviewer 3:

Authors designed a uniform nanoporous separator via in-situ photopolymerization of ZrMOC for the development of Li metal batteries. The Li symmetric cells with separator exhibit the better performance at a current density of 10mA cm^{-2} in the conventional carbonate-based electrolytes. The nano-structured pores in the separator was found to force partial desolvation of Li^+ and alter the reactivity between Li metal and electrolyte. However, some of the descriptions/experiments are questionable. Authors are encouraged to provide more experimental evidence and physical/chemical insights in the proposed mechanism.

We appreciate the reviewer's suggestions and comments.

Q1: Authors used excess Li (thick Li metal) which exceeds the capacity of NMC cathode in the Li metal batteries. Also, the vast Li sources leads to inflated Coulombic efficiency of the Li symmetric cells and decreases the energy density of the full cells. Thus, authors should examine the Li-Cu/Li-SS asymmetric cell to prove the significance of nanoporous separator in the carbonate-based electrolyte.

We appreciate the reviewer's suggestions, which helped us to improve the quality of our work. The Li | Cu asymmetric cells with PP separator and Zr-MOCN@PP were assembled to evaluate the Coulombic efficiency (CE) of the lithium metal anode in the carbonate-based electrolyte. The Zr-MOCN@PP based Li | Cu asymmetric cell showed a high CE of Li plating/stripping around 99.3%, which was significantly higher than PP separator-based cells (86.7%). Accordingly, we have added the sentences in the revised manuscript on **Lines 165-169**:

“Furthermore, the Li | Cu asymmetric cells with the PP separator and Zr-MOCN@PP were assembled to evaluate the Coulombic efficiency (CE) of the Li plating/stripping in the carbonate-based liquid electrolyte ($\text{LiPF}_6\text{-LE}$). The cell with the Zr-MOCN@PP achieved much high CE (99.3%), which outperformed the performance with the PP separator (86.7%) (Supplementary Fig. 22).”

The measurement information was added in the revised Supplementary Information:

“The Li plating/stripping Coulombic efficiency (CE) was evaluated by using Li | Cu asymmetric cells with the similar protocol in the previous report². Specially, the Li plating Q_1 (2mAh cm^{-2}) was used for Li plating/stripping (Q_2 , 1mAh cm^{-2}) for n cycles. A final Li stripping Q_3 exhausted all Li in the Cu current collector. The CE value was calculated as the following Equation S1:

$$\text{CE} = \frac{n Q_2 + Q_3}{n Q_2 + Q_1} \times 100\% \quad (1)$$

2. Yang, H. et al. Designing cation-solvent fully coordinated electrolyte for high-energy-density lithium-sulfur full cell based on solid-solid conversion. *Angew. Chem. Int. Ed.* **60**, 17726-17734 (2021).

The experimental data was added as Supplementary Fig. 22 in the revised Supplementary file:

Supplementary Figure 22 | Coulombic efficiency of Li plating/stripping. **a**, Li plating/stripping profiles and **b**, Coulombic efficiency in the PP separator and Zr-MOCN@PP based cells at a current density of 0.5 mA cm⁻² and 0.5 mAh cm⁻².

Q2: In the fundamental understanding, Li dendrites are formed at the Sand's time when the current density is over limiting current density. The formation of Li dendrites can be suppressed under low current density. Also, high Li⁺ transference number (t_{Li^+}) plays crucial roles in the Li plating-stripping process. However, authors only mentioned that MOF-modified separator exhibits higher Li⁺ transference number in the manuscript and did not provide the detailed discussion/experimental evidence regarding the limiting current/ Li⁺ transference number.

We appreciate the reviewer's suggestions, which helped us to improve the quality of our work. We have measured the Li⁺ transference number, and the result was added in the revised manuscript on **Lines 324-326**:

"It worth noting that the transference number (t_{Li^+}) of the Zr-MOCN@PP cell in this work was measured to be 0.69 (Supplementary Fig. 29), which is higher than the conventional liquid electrolyte system (~0.2-0.4)^{40,41}."

40. Zugmann, S. et al. Measurement of Transference Numbers for Lithium Ion Electrolytes via Four Different Methods, A Comparative Study. *Electrochim. Acta* **56**, 3926-3933 (2011).

41. Xu, K. *Nonaqueous Liquid Electrolytes for Lithium-Based Rechargeable Batteries.*

Moreover, the measurement information was added in the revised Supplementary Information:

“ Li^+ transference number (t_{Li^+}) measurement was carried out by using Li symmetric cell. A constant dc bias was set as 10 mV. Therefore, t_{Li^+} can be calculated from Equation S2

$$t_{\text{Li}^+} = \frac{i_1(\Delta V - i_0 R'_0)}{i_0(\Delta V - i_1 R'_1)} \quad (2)$$

In Equation 2, ΔV is the polarization voltage, and i is the current; the subscripts 0 and 1 indicate initial values and steady-state values, respectively, and R' is the sum of the charge transfer resistance and the passivating film resistance; R'_0 and R'_1 can be obtained from two impedance spectra on the cell in the frequency range between 0.1 Hz and 10 kHz before the polarization, and after the steady state has been reached.”

Supplementary Figure 29 | The transference number measurement. **a**, Impedance spectra of the Zr-MOCN@PP based cell with $\text{LiPF}_6\text{-LE}$ before polarization and after the steady-state has been reached. **b**, Polarization curve of the same cell.

Q3: Nanoporous separator has been used to suppressed the formation of Li dendrites and stabilize the Li plating-stripping process. Authors did not compare their results with other studies. The literature review is not completed in this work.

We appreciate the reviewer's suggestions, which helped us to improve the quality of our work. Accordingly, we added the references that reported the suppression of the formation of Li dendrite and stabilization of the Li plating/stripping process, and compared the results with these studies in the revised manuscript on **Line 140-143**:

“Although the suppression of the lithium dendrite by nano-porous separator have been reported²²⁻²⁷, inhibition the continuously occurred parasitic reaction between Li^0 and liquid electrolyte was firstly studied in this work.”

22. Li, Z. et al. *In Situ Chemical Lithiation Transforms Diamond-Like Carbon into an Ultrastrong Ion Conductor for Dendrite-Free Lithium-Metal Anodes. Adv. Mater.* **33**, e2100793 (2021).

23. He, Y. et al. *Simultaneously Inhibiting Lithium Dendrites Growth and Polysulfides Shuttle by a Flexible MOF-Based Membrane in Li-S Batteries. Adv. Energy Mater.* **8**, 1802130 (2018).

24. Wan, J. et al. *Ultrathin, flexible, solid polymer composite electrolyte enabled with aligned nanoporous host for lithium batteries. Nat. Nanotechnol.* **14**, 705-711 (2019).

25. Zhao, C.-Z. et al. *An ion redistributor for dendrite-free lithium metal anodes. Sci. Adv.* **4**, eaat3446 (2018).

26. Baran, M. J. et al. *Diversity-oriented synthesis of polymer membranes with ion solvation cages. Nature* **592**, 225-231 (2021).

27. Bai, S. et al. *High-Power Li-Metal Anode Enabled by Metal-Organic Framework Modified Electrolyte. Joule* **2**, 2117-2132 (2018).

Q4: The electrochemical impedance spectroscopy results of separators (PP and nanoporous separators) should be discussed during cycling.

We appreciate the reviewer's suggestions, which helped us improve the quality of our work. Accordingly, we have carried out the EIS measurements of the Li-symmetric cells with Zr-MOCN@PP and PP separator. The discussion was added in the revised manuscript on **Lines 132-136**:

“From electrochemical impedance spectroscopy (EIS) (Supplementary Fig. 12), before cycling (after 1 h of cell assembly), the Zr-MOCN@PP based cell demonstrated a slightly higher Ohmic resistance (R_{Ω}), but a significant lower SEI resistance (R_{SEI}) than the pristine PP separator based cell. And for the cycled cells, the R_{SEI} of Zr-MOCN@PP kept at a low value, while the R_{SEI} of PP largely increased from 10th cycle to 200th cycle.”

The detailed information for the impedance measurement was also added in the revised Supplementary Information:

“Impedance analyses of the Li-symmetric cells with PP separator and Zr-MOCN@PP were performed on a CHI660E Electrochemical Workstation (Shanghai Chenhua) with electrochemical impedance spectroscopy (EIS). The perturbation amplitude was 10 mV and the frequency range from 0.1 Hz to 10 kHz at room temperature.”

Supplementary Fig. 12 was added in the revised Supplementary file:

Supplementary Figure 12. Impedance spectroscopy of Li-symmetric cells with the PP separator and Zr-MOCN@PP before and after stripping/plating at a current density of 1 mA cm⁻², respectively. **a**, Before stripping/plating. **b**, **c**, after 10th, 200th stripping/plating of the PP separator and Zr-MOCN@PP based cells, respectively.

REVIEWERS' COMMENTS

Reviewer #1 (Remarks to the Author):

The authors adequately addressed Reviewer #1's concerns. Congratulations on the development of a compelling functional separator material.

Reviewer #2 (Remarks to the Author):

The author faithfully answered the questions and filled in the missing or ambiguous parts well. It seems that the core part of the polymerized Zr-MOC membrane study and the results supporting it have been sufficiently explained.

Reviewer #3 (Remarks to the Author):

The authors have addressed the questions and concerns properly.